

# Calibrating a wetland methane emission model with hierarchical modeling and adaptive MCMC

Jouni Susiluoto[1,3,4], Maarit Raivonen[2], Leif Backman[1,2], Marko Laine[1], Jarmo Mäkelä[1,4], Olli Peltola[2], Timo Vesala[2,5], and Tuula Aalto[1]

[1]Finnish Meteorological Institute, Erik Palmenin Aukio 1, 00560 Helsinki, Finland
[2]University of Helsinki, Department of Physics, Finland
[3]Lappeenranta University of Technology, School of Science, Finland
[4]University of Helsinki, Department of Mathematics and Statistics, Finland
[5]University of Helsinki, Department of Forest Sciences, Finland

*Correspondence to:* jouni.susiluoto@fmi.fi

**Abstract.** Methane ($CH_4$) emission estimation for natural wetlands is complex and the estimates contain large uncertainties. The models used for the task are typically heavily parametrized and the parameter values are not well known. In this study we perform a Bayesian model calibration for a new wetland $CH_4$ model to improve quality of the predictions and to understand the limitations of such models.

5    The detailed process model that we analyze contains descriptions for $CH_4$ production from anaerobic respiration, $CH_4$ oxidation, and gas transportation by diffusion, ebullition, and the aerenchyma cells of vascular plants. The processes are controlled by several tunable parameters. We use a hierarchical statistical model to describe the parameters and obtain the posterior distributions of the parameters and uncertainties in the processes with adaptive MCMC techniques. For the estimation, the analysis utilizes measurement data from the Siikaneva flux measurement site in Southern Finland.

10    The model parameters are calibrated using six different modeled peat column depths, and the hierarchical modeling allows us to assess the effect of the parameters on an annual basis. The results of the calibration and their cross validation suggest that the early spring net primary production and soil temperatures could be used to predict the annual methane emissions. The modeled peat column depth has an effect on how much the plant transport pathway dominates the gas transport, and the optimization moved most of the gas transport from the diffusive pathway to plant transport. This is in line with other research, 15    highlighting the usefulness of algorithmic calibration of biogeochemical models.

Modeling only 70 cm of the peat column gives the best flux estimates at the flux measurement site, while the estimates are worse for a column deeper than one meter or shallower than 50 cm. The posterior parameter distributions depend on the modeled peat depth. At the process level, the flux measurement data is able to constrain $CH_4$ production and gas transport processes, but for $CH_4$ oxidation, which is an important constituent of the total $CH_4$ emission, the determining parameter is 20    not identifiable.



## 1 Introduction

Methane is the second most important gas in the atmosphere in terms of its capacity to warm the climate, currently with the radiative forcing power of 0.97 $Wm^{-2}$. This is a sizable part of the total effect of well-mixed greenhouse gases, which is approximately 3.0 $Wm^{-2}$. According to IPCC (2013), the amount of $CH_4$ in the atmosphere has risen to its highest level in at

least the last 800000 years due to human activity, and based on ice core measurements, also its growth rate is presently very likely at its highest level in the last 22000 years.

    The sources of $CH_4$ are both anthropogenic and natural. In years 2003-2012, 60% of the global emissions were anthropogenic (range 50-65 %) and about one third came from natural wetlands. The most important source of uncertainty in the global methane budget is attributable to emissions from wetlands and other inland waters. Combining top-down and bottom-up

estimates, natural wetland emissions range from 127 to 227 Tg $CH_4$ $yr^{-1}$ (Saunois et al., 2016). Anthropogenic sources include rice paddies, landfills, enteric fermentation and manure, incomplete combustion of hydrocarbons, and natural gas leaks (Ciais et al., 2013).

    The methane from wetlands is produced by prokaryotic archaea under anaerobic conditions. The main sink for atmospheric $CH_4$ is its oxidation in troposphere by OH ions and the average lifetime of a $CH_4$ molecule in the atmosphere is $9.1 \pm 0.9$

years (Prather et al., 2012; IPCC, 2013).

    The wetlands in the boreal zone are a significant contributor to the total $CH_4$ emissions from wetlands (Kirschke et al., 2013), and for this reason the $CH_4$ emissions from them have been intensively studied, also with models, during the past years (Wania et al., 2010; Kaiser et al., 2016; Petrescu et al., 2015).

    The need for improved wetland methane emission modeling is amplified by the fact that although annual mean precipitation

is projected to increase in the boreal zone (Ruosteenoja et al., 2016), changes in the frequency and duration of severe drought may follow an alternate path (Lehtonen et al., 2014), manifesting the need to study wetland responses to extreme events.

    Changes to hydrological conditions such as draining or recurring low water table depth can alter the balance of greenhouse gas emissions (Frolking et al., 2011; Petrescu et al., 2015). Modeling and calibrating for such exceptional events can be difficult, as was found for instance by Mäkelä et al. (2016).

The HelsinkI Model of MEthane buiLd-up and emIssion for peatlands (HIMMELI) is a relatively full-featured wetland/peatland $CH_4$ emission model and it is described in detail in Raivonen et al. (2017). The model contains process descriptions for $CH_4$ production from anaerobic respiration, $O_2$ consumption and $CO_2$ production from oxic respiration, and gas transport processes via diffusion, ebullition, and plant transport. Modeling the concentrations of $CH_4$, $O_2$, and $CO_2$ in the peat column is explicitly included. The peat column depth can be set at any desired value, and the water table movement determines the part of the peat

column that is favorable for $CH_4$ production. The version of HIMMELI in this work has additional processes, described in Sec. 3.1, and the modified model is referred to as sqHIMMELI (square root HIMMELI), as it contains a description of $CH_4$ production from root exudates.

    Computer models describing environmental processes accumulate error at many levels (Sanso et al., 2007). The sources include time- and space discretization, incomplete physics and biochemistry descriptions, insufficient information about the





initial states of the model and their time evolution, numerical errors, and coding errors, along with parametrization-induced inaccuracies of the subgridsize processes. This leads to a need to calibrate and optimize models, as the physical variables do not necessarily correspond to the model variables perfectly and hence they cannot be often directly measured.

Several current $CH_4$ models include the important physical processes controlling both $CH_4$ production and transport in the peat column (Kaiser et al., 2016; Lai, 2009b; Müller et al., 2015). The modeled peat column depth affects the total modeled $CH_4$ emission from the peatlands and it is directly included in some models (Lai, 2009b; Walter and Heimann, 2000). These models are in general highly sensitive to changes in the values of the parameters (van Huissteden et al., 2009). However, even though algorithmic parameter optimization has been done in some studies, the stress is often on parameter efficiencies (van Huissteden et al., 2009), or optimal values (Müller et al., 2015), and hence the full uncertainty of the values of parameters in these models is not well understood.

Methane models typically use measured values from field campaigns and parameters estimated from those studies where applicable (Lai, 2009b; Walter and Heimann, 2000; Tang et al., 2010; Riley et al., 2011), and, when needed, include extra tuning parameters for processes (Walter and Heimann, 2000). This is a practical and much used route as information regarding all of the needed parameters is not available at all sites (van Huissteden et al., 2009; Walter and Heimann, 2000). Wide variability can be expected from some parameters, such as those controlling $CH_4$ oxidation (Segers, 1998). Emissions from different areas of the same wetland can also vary, due to microtopography and differences between how fast the peat decomposes in different areas (Lai, 2009a; Cresto Aleina et al., 2016), making straightforward parameter value assignment difficult.

Due to these uncertainties, values of parameters vary widely from research to research. For instance for the $Q_{10}$-value controlling the temperature dependence of $CH_4$ production, Walter and Heimann (2000) use the value 6, handpicking it from the interval of 1.7-16, whereas van Huissteden et al. (2009) use a range of 3-8, and Müller et al. (2015) constrain the value between 1 and 10, with the default value of 1.33 and eventually optimizing it to the value of 1 for two of the three optimizations presented. For other parameters, such as those controlling diffusion rates in peat, the situation is similar.

Calibration done for the models is usually quite basic. Wania et al. (2010) tune their model by running it with parameters from a parameter grid, containing only three values for each of the 7 parameters tested, and Riley et al. (2011) follow a similar procedure for the wetland $CH_4$ model component, CLM4Me, of the Community Land Model. Such crude sensitivity studies obviously are not able to find out how a model is able to perform at its best. Müller et al. (2015) have further optimized the CLM4Me model using an emulator combined with a simple minimization algorithm, with respect to several different sites, which are bound to have quite different physical characteristics, and are yielding optimal values often at the borders of the prescribed allowed area of variation. In a sensitivity analysis of the PEATLAND-VU model, a derivative of the Walter-Heimann model, van Huissteden et al. (2009) look at the efficiencies of the different parameters, but do not elaborate on other qualities of the posterior. Modeled peat column depth, which is included in the research at hand, is not optimized for the models generally, and for instance Walter and Heimann (2000) choose it based on expert knowledge.

Using hierarchical modeling to estimate annually changing parameters is sensible, since the flux measurement site has both properties that change from year to year (e.g. small changes in vegetation, plant roots, and microbe populations) and properties that are more permanent (e.g. peat quality and plant species). With fixed parameter values for all years, the model sometimes





does not accurately and appropriately describe the observations. On the other hand, with different parameters for all the years, the parameters are easily overfitted, meaning that while the resulting model fits the data well, it does not accurately predict future fluxes (Gelman et al., 2013). Hierarchical modeling provides a solution for these problems.

In the present study, the sqHIMMELI model is calibrated using adaptive Markov chain Monte Carlo (MCMC) techniques
to evaluate a hierarchical statistical model for the model parameters. The calibration is done for the boreal Siikaneva site. This study complements the work in Raivonen et al. (2017) in describing the effects of various parameters on the processes and fluxes, and analyzing what kinds of configurations best describe the studied boreal wetland.

Classical optimization is often misleading due to the multi-modality of the objective function, as for example Müller et al. (2015) reported in a study where they used a surrogate model to calibrate the parameters of the $CH_4$ model component of
the Community Land Model. This multi-modality can be accommodated for by using MCMC techniques. Utilizing MCMC methods for optimizing environmental models and studying their uncertainties is not new (Laine, 2008; Ricciuto et al., 2008; Hararuk et al., 2014), but to our knowledge they have not been used for wetland $CH_4$ model parameter estimation before. Moreover, the research that the authors are aware of does not investigate the interannual variability of parameters, as is done in this study.

The main objective of this work is to analyze the capabilities and limitations of a modern featureful wetland $CH_4$ model by looking into the shape of the posterior parameter distributions, parameter correlations, and the roles, identifiabilities, interdependencies, and interconnections of the parameters and the processes they control. The simulations and the analyses are performed with six different modeled peat depths, which allows for assessing how the modeled peat column depth affects the model behavior, and how deep a peat column is optimal based on the flux measurement data used.

## 20  2   Siikaneva wetland flux measurement site and model input data

Methane flux measurements were needed for estimating the model parameters, and for that purpose observational data from the Siikaneva peatland flux measurement site in southern Finland (61°50'N, 24°12'E) were used. The site is a boreal oligotrophic fen with a peat depth of up to four meters. The data collection started in 2005, and in this work eddy covariance (EC) $CH_4$ flux measurements from years 2005 to 2014 were used. In the current application of the EC method, the flux was calculated
from the wind speed and direction, and $CH_4$ concentration information, both of whose sampling frequency was 10 Hz. During the measurement period several different instruments were used for methane concentration measurements: Campbell TGA-100 (2005-2007 and 04/2010-08/2010), Los Gatos RMT-200 (01/2008-02/2014), Picarro G1301-f (04/2010-10/2011) and Los Gatos FGGA (2014). The wind velocity vector was analyzed by a USA-1 acoustic anemometer by METEK (Rinne et al., 2007). All the EC-data were post-processed in a consistent manner using an in-house software EddyUH (Mammarella et al.,
30 2016).

The required inputs for sqHIMMELI are daily soil temperatures, water table depths (WTD), net primary production (NPP), and leaf area indexes (LAI). The soil temperature profile for the grid used was generated by interpolating between the measurement depths (-5 cm, -10 cm, -20 cm, -35 cm and -50 cm) and assuming that at -3 meters and below the temperature is a constant





+7°C. This was the mean temperature of all the years at -50 cm depth. The WTD data used was available as measurement data, and where data was missing, it was gap-filled by repeating the previous measured value. Net primary production cannot be measured in a direct way, and hence modeled values for it were used. Also for LAI, a simple model was used for obtaining the input. For more details, see Appendix E. A summary of the data used is given in Table 1.

## 3 The sqHIMMELI model

The HIMMELI (HelsinkI Model of MEthane buiLd-up and emIssion for peatlands) model (Raivonen et al., 2017) is a detailed model for estimating $CH_4$ emissions from wetlands. It was developed at the University of Helsinki in collaboration with the Finnish Meteorological Institute and the Max Planck Institute for Meteorology in Hamburg. The model is designed to be used as a submodel for different larger modeling environments, such as regional and global biosphere models. It contains processes describing the production of $CH_4$ and $CO_2$, the loss of $CH_4$ and $O_2$, and transport of $CH_4$, $O_2$, and $CO_2$ between the soil and the atmosphere. The $CH_4$ transport can take place by diffusion in peat (in water and in the air), by ebullition (transport by bubble formation), and by diffusion in the porous aerenchyma tissues in vascular plants. The model is driven with peat temperature, WTD and LAI of the aerenchymatous plants. The process descriptions are mainly adopted from previous wetland $CH_4$ models such as Arah and Stephen (1998), Wania et al. (2010) and Tang et al. (2010). The version of the model used here differs slightly from that presented in (Raivonen et al., 2017), and is therefore called with the different name of sqHIMMELI to avoid confusion.

The model simulates the processes in a discretized peat column. The number and thickness of the peat layers can be varied, but in this work a variable number of 10 cm layers is used, similarly to e.g. Kaiser et al. (2016). Effectively, the total depth of the peat column changes, not the thickness of the layers. The water table divides the column into water-filled and air-filled parts, and $CH_4$ is produced only in the inundated anoxic layers. In the present configuration, the NPP-related $CH_4$ production is allocated into the layers according to the vertical distribution of the root mass, described in Sect. 3.2. The internal time resolution of the model is dynamically adjusted depending on the model state, and the output interval is set to one day.

HIMMELI itself, as presented in Raivonen et al. (2017), does not simulate carbon uptake (photosynthesis) or peat carbon pools but instead it takes as input the rate of anoxic respiration. The differences between HIMMELI and sqHIMMELI are described below in Sec. 3.1 and 3.2 and in Appendix A3.

For each modeled process in sqHIMMELI, there are parameters regulating the process, affecting the concentrations of $CH_4$, $O_2$ and $CO_2$ in the peat column, and the wetland methane emissions. The equations describing the physics relevant to the optimized parameters are listed in section 3.4. Other relevant model equations are listed in Appendix A.

### 3.1 Root exudates and peat decomposition

Methanogens prefer recently assimilated fresh carbon as their energy source, for instance the root exudates of vascular plants (Joabsson and Christensen, 2001). A connection between ecosystem productivity and $CH_4$ emission has been observed in several wetland studies (Bellisario et al., 1999; Whiting and Chanton, 1993). However, anoxic decomposition of litter and





older peat also produces $CH_4$ (Hornibrook et al., 1997). Many models form $CH_4$ substrates by extracting directly a fraction of the net primary production (van Huissteden et al., 2009; Wania et al., 2010), and some rely on heterotrophic peat respiration only (Riley et al., 2011). In sqHIMMELI both primary production and anaerobic peat decomposition were included.

The modified sqHIMMELI model contains an exudate pool description, from which it produces methane via Eq. 3 and A5.
The exudate pool itself is described by Eq. 4, detailing how the modeled NPP turns into root exudates. Effectively, a fraction of NPP determined by the parameter $\zeta_\mathrm{exu}$ produces root exudates, which are then distributed as anaerobic respiration according to the root distribution into the peat column at the rate determined by the model parameter $\tau_\mathrm{exu}$. The part ending up under the water table produces $CH_4$ and $CO_2$, depending on the oxygen content of the water, and above the water table the exudates are respired into $CO_2$.
The second source of anaerobic respiration, the anaerobic peat decomposition, is modeled in sqHIMMELI with a simple $Q_{10}$-model adopted from Schuldt et al. (2013). The peat under the water table is prescribed a turnover time, based on which anaerobic respiration and $CH_4$ are produced according to Eq. 5 and A6.

## 3.2   Root distributions

The sqHIMMELI model differs from HIMMELI in the details regarding the root distribution model. Compared to measurement
data of root distributions of aerenchymatous sedges from Saarinen (1996), the original root distribution $\pi(z)$, adopted from Wania et al. (2010) and described by

$$\pi(z) \propto \exp(-z/\lambda_\mathrm{root}), \tag{1}$$

does not describe the distribution of roots well. Here $z$ is depth, and $\lambda_\mathrm{root}$ is a parameter describing the steepness of the decaying exponential curve. This formula is replaced with

$$\pi(z) \propto C_0 \exp\left[-\frac{(z-z_0)^2}{\lambda_\mathrm{root}^2}\right] + C_1. \tag{2}$$

With the Gaussian shape, the new root density decreases faster with depth. Without this change, the optimization process calibrates the model to have very high root masses below 50 cm underground. The other difference between the models is that in the original model there are vanishingly few roots below the depth of one meter, but according to Saarinen (1996), sedge roots can reach to as low as 2.3 m under the surface. The term $C_1$ in Eq. 2 was added to remedy this.
Before starting the optimization, the parameters $C_0$, $C_1$, and $z_0$ were fitted to data from Saarinen (1996), resulting in values of $C_0 = 215$, $C_1 = 6$, and $z_0 = 0.105$. The different root distributions are shown in Fig. 1.

## 3.3   Peat depth

Methane is produced from anaerobic peat decomposition at all peat depths in the sqHIMMELI model, and its transport and oxidation affect the modeled $CH_4$ emission. The homogeneous model description of the peat column is highly idealized, as in
reality the peat column varies from place to place with respect to $CH_4$ production rate, production depth, and gas transport. Increasing peat depth in the model is a liability, since the deeper the column, the more expensive the model is to run (see Sect.





F2). The model calibration is run for the peat depths of 40, 50, 70, 100, 150, and 200 cm in order to find the optimal peat column depth for the model.

### 3.4 Parameter descriptions for sqHIMMELI

The parameters for the optimization were chosen to constrain the processes most important for the $CH_4$ emission. Of the optimized parameters, all but $\zeta_{\mathrm{exu}}$ and $Q_{10}$ are constant for all years. However, $\zeta_{\mathrm{exu}}$ and $Q_{10}$ change year to year to reflect the changes in the relative $CH_4$ input to the system from peat decomposition and NPP-based production. This will allow to analyze the year to year changes in relative importances of the production pathways. The setup is natural, as for example Bergman et al. (2000) report the $Q_{10}$-values changing from measurement date to another, even within a single year.

The parameters and their physical meanings are

**$CH_4$ production-related parameters**

1. $\tau_{\mathrm{exu}}$: Controls the decay rate of exudates, $\nu$, from the root exudate pool $P_{\mathrm{exu}}$,

$$\nu = \frac{P_{\mathrm{exu}}}{\tau_{\mathrm{exu}}}. \tag{3}$$

2. $\zeta_{\mathrm{exu}}$: Fraction of NPP carbon that goes to the root exudate pool.

$$\frac{\mathrm{d}P_{\mathrm{exu}}}{\mathrm{d}t} = -\nu + \psi_t \zeta_{\mathrm{exu}}, \tag{4}$$

where $\psi_t$ is the rate of NPP at time $t$, $P_{\mathrm{exu}}$ is size of the root exudate pool, and $\nu$ was given by Eq. 3.

3. $\tau_{\mathrm{cato}}$: Controls the base rate of peat decomposition into $CH_4$ in Eq. 5.

4. $Q_{10}$: Controls the temperature dependence of the rate of peat decomposition into $CH_4$ in anaerobic conditions via factor $k_{\mathrm{cato}}$, given by the equation

$$k_{\mathrm{cato}} = Q_{10}^{\frac{(T-273.15)}{10}} / \tau_{\mathrm{cato}}. \tag{5}$$

**Oxidation and respiration parameters**

5. $V_{R0}$: Respiration parameter controlling the rate of heterotrophic respiration, which consumes $O_2$ and produces $CO_2$. This affects the rate of temperature dependent heterotrophic respiration, $V_R(z)$, given by

$$V_R(z) = V_{R0} \exp\left(\frac{\Delta E_R}{R}\left(\frac{1}{283} - \frac{1}{T(z)}\right)\right). \tag{6}$$

Here $\Delta E_R$ is a parameter affecting the temperature dependence of the heterotrophic respiration, $R$ is the universal gas constant, and $T(z)$ is temperature at depth $z$.





6. $V_{O0}$: CH$_4$ oxidation parameter controlling the potential rate of CH$_4$ oxidation $V_O$:

$$V_O(z) = V_{O0} \exp\left(\frac{\Delta E_{\text{oxid}}}{R}\left(\frac{1}{283} - \frac{1}{T(z)}\right)\right).$$

(7)

Here $\Delta E_{\text{oxid}}$ is a parameter affecting CH$_4$ oxidation that is not part of the optimization.

**Gas transport-related parameters**

7. $\lambda_{\text{root}}$: Controls how the root mass is distributed. See Eq. 2.

8. $\rho$: Root-ending area per root biomass, affecting root conductance, see Eq. 8.

9. $\tau$: Root tortuousity parameter affecting the root conductance $K_R$. A tortuousity of 1 means that the roots are not decreasing the conductance via their curvedness. The equation for the conductance is

$$K_R(z) = \frac{D_{\text{air}} m \rho \pi(z)}{\tau z},$$

(8)

where $\pi(z)$ is the root mass density as a function of depth, over which the sum of the density is one, and $m$ is the total root mass per square meter, set to be proportional to LAI.

10. $f_{D,a}$: Fraction of the diffusion rate in air-filled peat divided by the diffusion rate in free air. The parameter affects the diffusion flux in the model: the higher this parameter is, the more there is diffusion as it takes a shorter time for the CH$_4$ to exit the peat reducing the possibility of oxidation and increasing the concentration gradient driving diffusion. The equation is

$$D_{\text{air}} = f_{D,a} D_{\text{air}}^{273} \left(\frac{T}{298}\right)^{1.82},$$

(9)

where $D_{\text{air}}$ is the diffusion rate in air-filled peat, $D_{\text{air}}^{273}$ is the diffusion base rate at $273K$, and $T$ is the temperature. This parameter is also present in Eq. 8.

11. $f_{D,w}$: Same as above, but in water. The equation describing the peat-water diffusion rate is

$$D_{\text{water}} = f_{D,w} D_{\text{water}}^{298} \frac{T}{298},$$

(10)

where the terms are analogous to the ones in the previous equation.

## 4   Model calibration and MCMC

The model calibration consisted of two steps: optimization and MCMC. Both of these steps were run separately for each different number of peat soil layers assessed (4, 5, 7, 10, 15, and 20 layers, each layer corresponding to 10 cm of peat). In the following, an *experiment* refers to one of these MCMC runs.



## 4.1 Calibration algorithms

For optimization, an initial parameter vector was first drawn from the prior, and the parameters were then optimized against the costfunction described in Eq. 11. The algorithm used was the simplex-based BOBYQA, described in Powell (2009). In our tests, it was significantly faster to converge than NEWUOA (Powell, 2004), L-BFGS or Nelder-Mead (Nelder and Mead, 1965). For each experiment, the model was optimized by running 350 model simulations with the minimization algorithm, which was enough for finding a local minimum to start the MCMC sampling from.

At the points obtained in the optimization, the model was linearized and from the Jacobian a suitable initial proposal covariance matrix for MCMC was estimated. After this the MCMC sampling was performed to estimate the posterior distribution.

In order to be able to assess the annual parameter and $CH_4$ transport pathway changes, a hierarchical description for two of the parameters was used. These parameters were $Q_{10}$ controlling the temperature dependence of the peat decomposition rate, and $\zeta_{\mathrm{exu}}$, regulating the production of root exudates from NPP.

The *hyperparameters* are the means and variances defining the Gaussian priors of the hierarchical parameters $Q_{10}$ and $\zeta_{\mathrm{exu}}$. They were updated using fixed Gaussian *hyperpriors* with Gibbs sampling. The sampling distribution depends on the current values of the hyperparameters. The role of the hyperprior is to constrain the distribution from which the hyperparameters are sampled.

Technically, a *Metropolis-within-Gibbs*-method Gelman et al. (2013) for sampling the hierarchical parameters, non-hierarchical parameters, and the hyperparameters was used, presented briefly in Appendix D. The model parameters (i.e. everything except the hyperparameters) were sampled with the Adaptive Metropolis (AM) MCMC algorithm (Haario et al., 2001), which uses a Gaussian proposal distribution, whose covariance matrix is adapted as the chain evolves, and over time the acceptance rate gets closer to an optimal value, which is 0.23 for Gaussian targets in large dimensions (Roberts et al., 1997). If the algorithm proposes values outside the hard parameter limits listed in table 2, the model will not be evaluated and the value is rejected.

Our empirical data for the hierarchical model were the nine years from 2006 to 2014, meaning that for each of these years there were corresponding $\zeta_{\mathrm{exu}}$ and $Q_{10}$ parameters in the optimization. The model needed to be spun up for each annual flux estimation in order to have a realistic column of gas concentrations available. For this reason, the previous year was always also simulated, and for the likelihood only the residuals from the latter year were included in the calculations. Therefore year 2005 did not contribute directly to the values of the objective function. The different years were run in parallel to save execution time.

## 4.2 Objective function

As in many practical MCMC applications, a major part of the parameter estimation problem is the proper definition of the objective function. It is defined here based on *a priori* information about the measurement uncertainties, based on information from the model residuals, and based on annual flux estimates. Additionally prior information about the parameter values is also utilized.





### 4.2.1 Model residuals and error model

The second component of the objective function contains the daily $CH_4$ flux measurements. It is assumed that the daily flux estimate uncertainty is dependent on a fraction $\alpha$ of the flux measurement (Richardson et al., 2006) and some constant error, $\gamma$ (e.g. measurement device precision). The model error is expected to follow a similar form, and hence $\alpha$ and $\gamma$ contain the contributions from both the model and measurement errors.

The combined error is described by a Laplace distribution. The flux observations are reported to follow a distribution of this type, rather than a Gaussian distribution (Richardson et al., 2006).

When determining the parameters $\gamma$ and $\alpha$, the resulting residuals end up being autocorrelated. Therefore they are treated as such with the AR(1)-model, described e.g. in Chatfield (1989). Applying it, a set of Laplace-distributed residuals $r^*$ is obtained. The error model is explained in more detail in Appendix B.

### 4.2.2 Annual $CH_4$ fluxes

For future climate projections, the annual total emission of the snow-free period is one of the important quantities, as cumulative emissions are what determine the radiative forcing. Therefore a Gaussian-distributed observation, $G_{\mathrm{obs}}$, is included for the total annual flux, with error variance $\sigma_G^2$, where $\sigma_G$ is set to 10% of the annual total flux for each year. The modeled total annual flux is denoted by $G_M$. This term keeps the annual emission estimates reasonable in the early stages of the sampling. As the annual flux estimate errors are small when sampling parameters close to the posterior mode, this term has only a minor effect at the late stages of the MCMC.

### 4.2.3 Prior information

The parameters affecting the $CH_4$ production of the wetland model are not known well, but despite this, not setting any prior distributions on parameters can lead to nonphysical parameter values in the posterior distribution.

The parameter priors are set to zero outside prescribed bounds. Within these bounds, most parameters are assigned Gaussian priors, and for the others the priors are set to be flat. The prior values are based on both literature and expert knowledge and the information regarding the parameter values is summarized in Table 2.





### 4.2.4 The objective function

The *objective function* for the parameter optimization, $J(\boldsymbol{\theta})$, is the negative logarithm of the value of the posterior probability density function at $\boldsymbol{\theta}$. It combines our statistical knowledge of flux observations, annual flux estimates, and parameter priors presented in Sec. 4.2.1 – 4.2.3, and is given by:

$$J(\boldsymbol{\theta}) = -\log\left(p(\boldsymbol{\theta}|\boldsymbol{y})\right) =$$

$$\sum_{t=1}^{N_{\text{obs}}} |r_t^*| + \frac{1}{2}\left( \sum_{y=1}^{N_{\text{years}}} \frac{(G_M - G_{\text{obs}})^2}{\sigma_G^2} + \sum_{i=1}^{N_{par}} \frac{(\theta_i - \mu_i)^2}{\sigma_i^2} \right). \tag{11}$$

Here $|r_t^*|$ are the AR(1)-transformed Laplace-distributed residuals, $G_M$, $G_{obs}$ and $\sigma_G^2$ are the components of the annual flux term, and the last term is the prior contribution, where $\theta_i$ is the proposed parameter value, $\mu_i$ is the prior mean, and $\sigma_i^2$ is its variance. For technical details, see Appendix B.

## 5 Results and discussion

The experiments yielded an MCMC chain for each modeled peat depth and the final number of model simulations varied from 78000 to 391000. To look at statistics, 50% of values from the start of each MCMC chain were discarded as warm-up. The posterior covariance structures of the chains were found mostly to be similar to each other. The posterior distribution from the experiment with 70 cm of peat is shown in Fig. 2, and the correlation features for all peat depths are shown in the upper triangle of that figure. For the different processes, Fig. 11 shows an example of the posteriors and the process correlations.

For each MCMC chain, three different estimates for the parameters and fluxes were looked at: the maximum a posteriori ("MAP") estimate, posterior mean estimate ("PM"), and a "non-hierarchical" posterior mean estimate ("NHPM"), where the mean values of the parameters $\zeta_{\text{exu}}$ and $Q_{10}$ over the different years were used.

### 5.1 Parameter values and modeled peat depth

The parameter values of the MAP and PM MCMC optimizations are shown in Table 3. The catotelm carbon pool turnover time, $\tau_{\text{cato}}$ grows with the peat depth, and the reduction factor for the diffusion coefficient in air-filled peat, $f_{D,a}$, also grows slightly, increasing the diffusive permeability of the dry part of the column hence increasing conductance. The root conductance gets larger with the increasing peat depth, by the influence of the parameter $\rho$, which grows slightly, and the decrease in root tortuosity given by parameter $\tau$.

The MAP estimates of the different experiments disagree on the value of the water-diffusion rate coefficient $f_{D,w}$, and the posteriors shown in Fig. 3 (k) are wide, especially for the experiments with less peat, meaning that especially in those cases this parameter is highly uncertain. However, with increasing peat depth, the mass of the posterior distribution mass moves closer to 1 compensating the decreased conductance caused by the longer distance to the surface. The air diffusion rate coefficient $f_{D,a}$ shows similar behavior, but with lower values as it is constrained by the prior. For the 40-cm peat column optimization, the



MAP estimate for $f_{D,w}$ is far from the others. This can be explained by that in that particular case there is not much water-filled peat in the model leaving the parameter with less effect on the results.

The root distribution parameter, $\lambda_{\text{root}}$, is optimized larger than expected, and is closer to the prior value only in the optimization with 15 and 20 layers. This is also true for the MAP estimates implying that the model optimizes best when the CH$_4$ produced from the photosynthesis-induced exudate production goes relatively far below the surface: with a value of 0.3, 49% of the roots are deeper than 25cm, 15% of the roots are deeper than 50cm, and just 2.5% are deeper than 75cm, see Fig. 1. In relation to these numbers, the water table depth is most of the time above the depth of -20 cm. Additionally, a larger $\lambda_{\text{root}}$ will facilitate the emission of the CH$_4$ produced by peat decomposition in the catotelm. The small values for the two experiments with the thickest peat column make the model behave differently from how it functions with 40-100 cm of peat.

The results in Table 3 reveal that the parameter regulating the exudate pool turnover time, $\tau_{\text{exu}}$, slightly decreases with the peat column depth implying a shorter period between photosynthesis and methane emission with more peat. However, $\overline{\zeta_{\text{exu}}}$ controlling the amount of methane produced from exudates gets smaller until 100 cm, and the values for 150 and 200 cm of peat are markedly larger. This implies, that CH$_4$ production from exudates is closely linked to the depth of the root mass and $\lambda_{\text{root}}$.

The non-hierarchical parameter $V_{O0}$ controlling the amount of CH$_4$ oxidation taking place does not show a trend with respect to the modeled peat column depth in the PM estimate, but there is a clear trend in the MAP estimates, shallow peat depths favoring larger parameters and inducing more CH$_4$ oxidation. The effect of the parameter on the total CH$_4$ oxidation is substantial, which is evident from part (a) of Fig. 4 With all peat depths, the chains traverse in regions of both high and low $V_{O0}$ as shown in Fig. 3 (f). Another parameter indirectly affecting CH$_4$ oxidation, the heterotrophic respiration parameter $V_{R0}$, drifts in all experiments close to its minimum value reducing the amount of heterotrophic respiration taking place.

Table 3 shows that the hierarchical parameter $Q_{10}$, controlling the temperature dependence of the CH$_4$ production from peat decomposition, increases with peat column depth: the more peat there is, the stronger the peat decomposition process responds to soil temperature changes. Contrasting with this, the parameter $\tau_{\text{cato}}$ controlling the peat decomposition rate in the catotelm increases with peat depth compensating for changes in total peat volume and keeping the production volumes reasonable. Part (d) of Fig. 4 shows how increasing the peat depth and keeping parameters constant drastically increases the total methane production.

The annual variability of $Q_{10}$ is similar across all peat depths. For instance years 2013 and 2014 (and sometimes 2011) are years of high $Q_{10}$, whereas in 2006 the parameter gets its lowest or second-lowest value for all depths in the PM simulations. For the other hierarchical parameter, $\zeta_{\text{exu}}$, these patterns do not exist.

## 5.2 Costfunction values and annual discrepancies

The minimum costfunction values and annual biases provide information about how well the different configurations of the model performed in the model calibration task. Table 3 lists the costfunction values for the MAP estimates, and the annual errors for the MAP, PM, and NHPM estimates are shown for each MCMC experiment in Fig. 5. Among the MAP estimates, the costfunction value is lowest with 7 layers of peat (259) and then gradually higher for 10 (262), 15 (267), 20 (269), 5 (274),





and 4 layers (295), the last of these being significantly worse than the first ones. Figure 5 suggests that for the MAP and PM estimates, the annual total CH$_4$ flux estimates are less steady with only 40 or 50 cm of peat. In figure Fig. 5 (g) the NHPM estimate shows a very large variance of the annual errors, with early years having a sizable positive bias, and later years having a similar negative bias. Incidentally the averages over the whole period for NHPM are small as Fig. 6 (b) indicates. The model

estimates of the annual fluxes are good and the variance of its errors is small for both MAP and PM experiments, especially for peat column depths of at least 50 cm. Compared to the default parameters, the estimates are much improved. This is to be expected as the results shown are not for an independent validation dataset. Rather, the motivation is to see, how the model fit looks like for optimized parameters and how the features differ from the unoptimized ones.

A cross validation of the annual errors is shown for the experiments with 70 cm and 100 cm peat columns in Fig. 6 (g) and 5

(b). While the annual estimates are worse than the estimates with the optimized parameters, compared to the NHPM estimate and the default parameter values the errors are smaller. Additionally the overall biases are as good as with the optimized parameters. The cross validation is described in Sec. 5.7.

Almost all years of hierarchically optimized experiments show at least a small negative annual bias when compared to the available observations. This can be due to the high day to day variability of the summertime fluxes, which dominate year-

round total fluxes, and the fact that the model can not, without data about the fine structure and heterogeneity of the wetland, match the high variability fluxes. The proportional model-data residual error component $\alpha\, y_t$ (Appendix B) allows the model to underestimate the high peaks more than the low flux values. The error model favors the baseline of the lower values during periods when observed variance is very high, for instance in the peak emission season of 2010. This is also true for periods of increased ebullition, and such fluxes are very difficult to fit into.These periods contribute to both the costfunction values and the

underestimation of the total flux. Any temporal shifts of peaks of seasons are penalized heavily, and the optimized parameter values rather produce less peaks than right size peaks at a slightly wrong time.

### 5.3    Parameter values and processes in sqHIMMELI

The sqHIMMELI model produces the CH$_4$ from anaerobic respiration that originates from peat decay and the decay of root exudates. These production components, along with the different output pathways, CH$_4$ oxidation and model residuals, are

plotted as functions of water table depth in Fig. 8 for the optima of various MCMC experiments, and for the unoptimized default parameter values. The process correlations and covariances are shown for the year 2008 from the experiment with 70 cm of peat in Fig. 11.

In the following, *all ebullition* refers to any ebullition in the peat column regardless to whether the bubbles reach the peat column surface. *Ebullition* refers to the part of "all ebullition" which reaches the surface. Most of the time the water table

is under the peat surface, and at those times "ebullition" is zero, although "all ebullition" can be substantial. In that case the ebullition flux does not go directly into the atmosphere, but into the first air-filled peat layer above the WTD, and continues from there via other pathways. The reason for this separation comes from implementation details of HIMMELI. In all experiments, ebullition reaching the surface is minor fraction of the total CH$_4$ emission.



For the PM estimate with the 100 cm column depth, the flux components and oxidation are shown as time series in Fig. 9. Having only four layers of peat leads to peat decay being inhibited when the water table is low as the volume of the modeled catotelm decreases (Fig. 8 (f)). The effect is seen also in the plant transport component in Fig. 8 (b). Plant transport becomes proportionally more important with increasing depth of the peat column with MAP, PM, and NHPM estimates (Fig. 5 (a-c)), even though the differences get quite small and the system seems to mostly stabilize already at 7 layers. For the default parameters, however, the trend is opposite (Fig. 5 (d)) as increasing the peat depth dramatically increases $CH_4$ production and as the default parameter set favors ebullition and diffusion over plant transport.

Comparing results from simulations with optimized parameters to results using the default parameter values (prior mean values, shown in Table 2) shows that the optimization drastically increases the role of the plant transport pathway at the expense of the diffusion pathway. Diffusion and all ebullition fluxes are closely tied to each other, as can be seen in Fig. 5 (a-d), in that in all cases their values are close to each other. This is also visible in the flux component time series in Fig. 9.

### 5.3.1  Methane production and oxidation

Figures 4 and 11 show, that there is considerable annual variation in the production of $CH_4$ from both of the production processes. Year 2007 has a high amount of production from peat decomposition, whereas year 2006 shows a lot less, even though the $\zeta_{exu}$-controlled proportion does not change much. This is not a general trend, though, and instead in years of high emissions the amount of $CH_4$ from both of the production sources is increased. The shape of the NPP input, shown in Fig. 7, does not change remarkably from year to year, but the emissions change considerably, as the model state and input affect the production non-linearly. For example in times of low WTD in the peak emission season, the root exudates do not contribute to $CH_4$ production, as much of the roots are located in the dry part of the peat column and the exudates are deposited there. Another explanation for changes in $CH_4$ production comes through the production-determining parameters, whose variation is in Sect. 5.7 found to be related to the springtime temperature and NPP.

The NPP-based $CH_4$ production controlled by the parameter $\zeta_{exu}$ is constrained by its hyperprior as can be seen in Fig. 3 (b) and the values accepted in the chain are from the higher side of the Gaussian hyperprior, whose mean and standard deviation are both 0.2. The posterior means in table 3 are between 0.27 and 0.38, with standard deviations of 0.17-0.18. For the MAP values the deviations are even higher, leading to wider fluctuations in the characteristics of the modeled wetland. As mentioned in Table 2, the prior for $\zeta_{exu}$ was set quite low, and actually even these values obtained here are on the low side of the spectrum reported by Walker et al. (2003), who gives a range of 0.2-0.84 in terms of our $\zeta_{exu}$-parameter. Our result hence agrees with that a relatively large portion of the photosynthesized sugar is respired into methane.

The methane produced by the action of $\zeta_{exu}$ is distributed according to the root distribution, whose form is determined by $\lambda_{root}$. The posterior means reveal, that that the contribution of the prior component of $\lambda_{root}$ to the costfunction is large. Its values might well be larger with a wider prior and more permissive prior, but in regard to how root distributions are in reality (Fig. 3.2), larger values for the parameter would make its interpretation difficult. This parameter affects both how exudates are allocated in the column and how deep the fast plant transportation reaches. Clearly there is a need to reach further down, implying that the model performs more optimally when it transports $CH_4$ faster to the atmosphere.



The methane production from decomposition of peat in anaerobic conditions is aided by the rather strongly correlated parameters $Q_{10}$ and the catotelm carbon decay half-life $\tau_{\text{cato}}$. Unlike with $\zeta_{\text{exu}}$, the prior means of $Q_{10}$ are mostly inside the 1-$\sigma$ bounds of the hyperprior, and the temperature dependence of the anaerobic respiration from peat decomposition is close to what was *a priori* expected.

All years of the PM simulations have very little oxidation taking place with over 70 cm of peat and in the experiments with up to 70 cm of peat the amount of oxidation is higher (Fig. 4 (b)). In our analysis no easy explanation was found for this feature and it is suspected that multiple processes involving the parameters governing the root distributions and the availability of oxygen are behind the phenomenon. The MAP simulations (Fig. 4 (a)), however, show that the CH$_4$ oxidation reduces gradually with the modeled peat depth, and that the simulations with the deepest modeled peat columns have parameter estimates with the

lowest $V_{O0}$-parameters. The correlation between oxidation and the value of $V_{O0}$ is high, at 0.8 for the year 2008 with 70 cm peat. With more modeled peat, the methane transportation time from the lower parts of the catotelm increases, and to compensate for this $V_{O0}$ in the MAP covaryingly reduces CH$_4$ oxidation so that the amount coming to the surface varies only a little (Table 3). This is also supported by that methane oxidation and production from root exudates covary negatively, as shown in Fig. 11. That figure also shows, that the exudate and peat decomposition based methane production terms are strongly

negatively correlated, and that either of the terms can dominate production of CH$_4$ within the 50% confidence interval, even though in 2008 the 70 cm experiment shows overall dominance of the peat decomposition process.

The production and oxidation related parameters $\tau_{\text{cato}}$, and $V_{O0}$ correlate (Fig. 2), and $\zeta_{\text{exu}}$ and $Q_{10}$ are affected via their correlations with $\tau_{\text{cato}}$. These parameters covary producing a total emission that minimizes the likelihood but this yields a posterior, where some parameters like $V_{O0}$ have wide marginal distributions (Fig. 3 (f)), as in the presence of several covarying

parameters any of those covarying ones can to some degree compensate for the movement of the others. The prior bounds of $V_{O0}$ were tight and for example Segers (1998) reports that potential CH$_4$ oxidation can vary across three orders of magnitude. Hence, higher proportions of CH$_4$ oxidation could have been seen with a more permissive prior. This would have then resulted in wider posteriors also for the covarying parameters.

Curiously, parameter $V_{R0}$ controlling heterotrophic respiration correlates negatively in all experiments weakly with CH$_4$

production via $\zeta_{\text{exu}}$, and positively with parameter $\tau_{\text{cato}}$. Removing oxygen from the column reduces CH$_4$ oxidation and in order to maintain the overall level of CH$_4$ emission, production is reduced. The wide posterior of $V_{R0}$ in Fig. 3 implies that the day to day variation in the emissions and the combined effects of other parameters dominate. In the MAP estimates, however, the parameter is close to the lower bound.

### 5.3.2 Plant transport

The amount of plant transport in the calibrated models, shown in Fig. 5, is close to 90% which is slightly over the upper end of the range of 68-85% reported in Wania et al. (2010) in a study simulating CH$_4$ emissions for seven boreal peatlands. This is opposite to what was obtained with sqHIMMELI with the default (prior mean) parameter values and 100 cm modeled peat, where the simulation routed 71% of the flux via diffusion.





The high optimized share of plant transport is due to the deep roots and high root conductance from the high values of the root depth controlling parameter $\lambda_{\mathrm{root}}$ and the root ending cross section parameter $\rho$, and the low values of root tortuousity parameter $\tau$. These parameters are close to the limits of what the priors allow, and are the reason for that plant transport dominates the gas transport. Wania et al. (2010) used the parametrization from Eq. 1 with $\lambda_{\mathrm{root}} = 0.2517$, and the root distributions

from the PM estimates are shown alongside that distribution in Fig. 1. Compared with measurements from Saarinen (1996), the amount of roots at 20-60 cm is exaggerated by all of the optimized parameter values. The model provides a better fit to the data when the root conductance is maximized.

The parameter posteriors of $\lambda_{\mathrm{root}}$ in the MCMC with 15 and 20 peat layers are apart from the others in Fig. 3, implying that an optimal rooting depth is an ambiguous notion. The root distribution depths also correlate differently with other parameters

in different depths - there is a negative correlation between $\overline{\zeta_{\mathrm{exu}}}$ and root depth (Fig. 2), which gets stronger with increasing peat column depth suggesting that more exudates are needed for shallow roots, which is reasonable since the exudates above the water table are respired aerobically. The other parameters affecting plant transport, $\rho$ and $\tau$ both are included in Eq. 8 and one could expect them to be tightly coupled. In the posterior, however, they are only slightly correlated, with coefficients in Fig. 2 from 0.16 to 0.31. The strict priors may play a role, as root tortuousity cannot go below the value of 1.

### 5.3.3  Diffusion

The masses of the diffusion coefficient parameters $f_{D,a}$ and $f_{D,w}$ in the posterior distributions (Fig. 3 (j) and (k)) are above the priors. This is true especially for $f_{D,w}$, which optimizes to close to the upper limit of one, specifically for the experiments with 100 - 200 cm of peat. Kaiser et al. (2016) note that these parameters are not well known, and use for both of them the value of 0.8, in which light the prior for $f_{D,a}$ looks narrow. The PM estimates for $f_{D,a}$ in Table 3 are between 0.50 and 0.65

for the depths of 40 - 100 cm.

The parameter $f_{D,a}$ is correlated negatively with the root-ending area parameter $\rho$. This is because the air diffusion parameter also affects the speed of $CH_4$ transport in plant stems via Eq. 8, and by negatively correlating the two parameters the model can compensate for one of them by moving the other. Additionally, a smaller root conductance implies that more of the $CH_4$ needs to come out via the diffusive flux, which is also seen in the negative correlation of $\lambda_{\mathrm{root}}$ and $f_{D,w}$, especially in experiments

with peat depths of 70 and 100 cm.

Diffusion is correlated strongly with peat decay-based $CH_4$ production and negatively with exudate-based production (Fig. 11), and these correlations extend to the hierarchical parameters defining the $CH_4$ production (not shown). This is related to the strong connection between diffusion and ebullition, and that decaying peat produces $CH_4$ lower in the peat column than decaying exudates, production from which is more likely transported by plants (Fig. 11).

In general, the calibrations tend to end up facilitating the total $CH_4$ transport as the depth increases, by the action of the parameters $\rho$ and $\tau$ affecting plant transport, and $f_{D,a}$, and $f_{D,w}$ affecting diffusion, implying that there is a regime of optimal conductance. Pertaining to this, Fig. 6 shows how the more modeled peat there is, the less important the diffusive component becomes. Going deeper down, plant transport becomes more competitive compared to diffusion in the MAP estimates.



### 5.3.4 Ebullition

Ebullition is very strongly tied to diffusion in the flux estimates with parameters from the posterior, as is shown for the 70 cm experiment in Fig. 11. The flux component timeseries in Fig. 9 shows that ebullition to the surface is a small fraction (circa 0-3% with optimized parameters), of the total flux, and Fig. 6 shows, that the more there is peat, the less important the

ebullition flux is, including the part emitted as part of the diffusive flux. Similarly, Wania et al. (2010) report almost virtually no ebullition to the surface. This result is highly dependent on the type of the wetland as for instance Kaiser et al. (2016) report high ebullition fluxes for a polygonal tundra in the Siberian permafrost region, where the ice-free soil layer reaches only about 30 cm depth during summer. Variation between different sites is very large and depends on whether the water reaches the surface at times of high $CH_4$ emission.

Contrasting with this, in the simulations with the default parameters and 100 cm of peat, a major part of the diffusive flux is transported by ebullition (Fig. 6) and diffusion is the dominating flux component, even though ebullition to the surface accounts for only 5% of the total flux. Since ebullition is a fast timescale process, it was not directly constrained in the optimization with parameters, as preliminary tests revealed that daily data resolution would not be sufficient for this. While finer time resolution data would have been available, using it would not have been feasible as there is not enough knowledge about the fine structure

of the wetland and micrometeorological conditions affecting the footprint area of the flux tower. It is reasonable to believe that the deviations from the daily averaged fluxes at a finer time resolution would only look like noise in the residuals not improving our parameter posterior. Despite this, ebullition is controlled indirectly by letting $CH_4$ production and transport parameters control when the water column has enough $CH_4$ available for ebullition. This happens when the sum of the partial pressures of dissolved gases is larger than the sum of atmospheric and hydrostatic pressures as shown in Eq. A13. The high

ebullition-related proportion of the diffusive flux strengthens the argument that the likelihood formulation results in model optimizing towards parameter values that support rapid $CH_4$ transport.

The results show that with deep roots and high root conductances the wet part of the peat column rarely creates the conditions for ebullition to happen. Hence with less peat the amount of "all ebullition" increases, (Fig. 5 (b), 6 (a), and 8 (d)), as the produced $CH_4$ needs to be stored in a smaller volume increasing its concentration. This way the modeled peat depth has a

major effect on how the model transports gases.

### 5.4 Parameter and process identifiability

The priors of the hierarchical $CH_4$ production-related parameters $Q_{10}$ and $\zeta_{exu}$ in Fig. 3 (b) and (d) are constrained by the data, as are the hierarchical parameters themselves, shown in Fig. 10. The priors of these distributions are wider than their posteriors, which is also the case for the other production-related parameters $\tau_{exu}$ and $\tau_{cato}$. The different posteriors of the catotelm peat

turnover time $\tau_{cato}$ are disjoint from each other as increasing simulated peat column depth is compensated for by reducing the peat decomposition rate per volume. Both process descriptions for obtaining the anaerobic respiration are clearly needed for a good model fit, because the parameter posteriors do not have remarkable mass in the regions minimizing either of these processes (hierarchical parameters at the lower bounds or turnover rate parameters $\tau_{exu}$ and $\tau_{cato}$ at the upper bound). The





covariances in Fig. 11 and Fig. 2 show that the two production processes covary, and hence they are partly interchangeable. Reasonable identifiability of the $Q_{10}$-parameters is not obvious, as for example Müller et al. (2015) optimizing a corresponding parameter end up with the parameter at the lower bound of their prescribed range.

The posterior distributions of $V_{R0}$ show, that sqHIMMELI performs better when the heterotrophic respiration is close to
being minimized, but still away from the lower bound. With 100 cm of peat, the parameter has a clear mode further from the lower bound, suggesting that the flux measurement data used also constrains this process, and that the prior does not indisputably rule out the best values. However, the oxidation parameter $V_{O0}$ is not identifiable, and as the strong correlation with the peat decay related parameter $\tau_{\mathrm{cato}}$ shows, its function is partly to calibrate the total CH$_4$ output of the the model and to spread the posteriors of the covarying parameters. With this model and data, methane oxidation rates at the Siikaneva site
cannot be estimated without further constraints for e.g. the CH$_4$ production.

All the parameters affecting root conductivity are constrained by the data to maximize the conductance. The root tortuousity parameter $\tau$ has narrow posteriors close to the lower bound of one, the root depth parameter $\lambda_{\mathrm{root}}$ is above its prior, and the root-ending area parameter $\rho$ optimizes to very high values compared to the prior distribution (Fig. 3 (g-i)). The diffusion-related parameters $f_{D,a}$ and $f_{D,w}$ are optimized to high values and identifiable with the exception that with the shallowest peat
depths the water diffusion rate coefficient has little role and a wide posterior spanning all values from 0 to 1. Transport pathway shares are stable between the MAP and PM optimizations in Fig. 6 (a), and their annual variation is small, implying that the existence of the different pathways helps to optimize the model fit.

## 5.5 Low WTD in 2006, 2010, and 2011

The sqHIMMELI model is not able to estimate the CH$_4$ flux correctly in times of low water table in the 40 and 50 cm peat depth
configurations. This is not unique to this particular model - also other studies have indicated the challenges in parametrizations of emission models in response to the water table depth (e.g. Zhu et al. (2014)). Figures 8 (b) and (f) reveal that in July 2006 the plant transport component and CH$_4$ production from peat decay go effectively to zero during the period of the lowest WTD. This effect is not visible in the simulations with more peat, and already the PM estimate with 7 layers gives a very nice fit for July 2006 (not shown) as the deeper peat column provides for more freedom for flux adjustment. The underestimation of the
emission is visible comparing the plant transport pathways and total model residuals with respect to the water table depth at different depths in Fig. 8 (b), (f), and (h).

Other extended periods of low water table occur during the years 2010 and 2011, which explains why those years tend to be accentuatedly underestimated with respect to the observed flux with shallow simulated peat columns, as is shown in Fig. 5 (f), even though curiously the 50 cm MAP estimate also performs well. The lowest water table depth of the simulation period is in
2006, when on the 26$^{\mathrm{th}}$ of August the water table drops to 38 cm below the surface. In 2011 the WTD goes below 20 cm for a total of 77 days in a row, and in 2010 it recedes to -17 cm for a period of 71 days, the average of the period being -23 cm. Years 2010 and 2011 have the strongest tendency to underestimate the total annual CH$_4$ emission compared with the observations. A sufficiently deep peat column to accommodate for CH$_4$ emissions during the low WTD periods is needed for making accurate predictions .



### 5.6 Optimal modeled peat depth for sqHIMMELI

Even though most of the parameters and processes are identifiable, all of the parameter posteriors vary with peat depth, the most striking example of which is $\tau_{\mathrm{cato}}$ (Fig. 3 (c)). For this reason, the validity and meaning of the parameter values must be understood in each particular model setting.

The objective function incorporating prior knowledge can be used to evaluate what peat column depth best represents the data and still retains the physical interpretation of the parameters, information about which is in the prior parameter distributions. In the MAP estimations, the costfunction values (Table 3) and the annual flux estimate errors (Fig. 6) are smaller starting with the depth of 70 cm and especially the 40 cm optimizations are systematically worse in this respect than the others, due to worse handling of periods of low WTD (Fig. 8). These problems do not exist with the 70 or 100 cm simulations, and are less pronounced already with 50 cm of peat.

With 150 or 200 cm of peat the correlations of the parameters shown in Fig. 2 show markedly different patterns from the correlations with shallower modeled columns. For these thick peat columns the costfunction values are higher, the correlations are not easy to explain, annual negative biases are not better, and model integration is more costly in terms of CPU time. For these reasons there is no reason to believe that modeling deeper peat columns than 100 cm in sqHIMMELI would be superior. Rather, the optimal thickness lies between 50 and 100 cm.

### 5.7 Predicting emissions with sqHIMMELI

Modeled $CH_4$ flux estimates may have large errors as was shown in Fig. 6 (b) with the default parameter set. The negative biases of less than ten percent in the calibration phase that were found with the PM estimates are reasonable since the quality of the modeled input data from e.g. a land surface scheme will also contribute to the uncertainty in the model predictions.

Compared to the estimate with the optimized annual variations of the $Q_{10}$ parameters, the posterior mean estimate without the hierarchical parameters (NHPM) does not produce very good good flux estimates over the assessment period (Fig. 5 (g)). With all peat depths, the total $CH_4$ emission of the first years is overestimated by up to 30 percents and for the last years there is a similar negative bias. The hierarchical posterior mean (PM) on the other hand does produce very steady estimates of the $CH_4$ flux, compared with observations, and for these estimates the model dynamics are similar between the estimates for the peat depths of up to 100 cm.

In order to be able to utilize the PM estimates for the future prediction of $CH_4$ emissions, the values of the hierarchical parameters need to be estimated for the simulation years. A simple regression analysis of the hierarchical variables with respect to relevant input data was performed in order to find out if such estimation is possible. As the explaining variables, means, minimums, and maximums of NPP, water table depth, and soil temperature at different depths and over different periods of time were looked at. These time periods were June, July, August, and various different amounts of days from the start of the year.

The analysis revealed that the mean soil temperature of the first 10 weeks of the year at the depth of 30-40 cm, denoted here by $\overline{T_{30-40}^{70}}$, is the best single-variable predictor of the $Q_{10}$-value for that year, and for $\zeta_{\mathrm{exu}}$, it is the sum of NPP from the first





130 days of the year, denoted by $NPP^{130}$. This is hardly surprising, since the peat decomposition process regulated by the parameter $Q_{10}$ is driven by soil temperature, and the anaerobic respiration from exudates controlled by the parameter $\zeta_{\mathrm{exu}}$ is driven by the NPP input.

The $p$ values summarizing the reliabilities of the regressions and the $r^2$ values, which are the coefficients of determination of the fit, are presented in table 4. The $r^2$ values explain what fraction of the variance of the dependent (predicted) variable is explained by the independent (explaining) variables.

For the MCMC experiment with 40 cm of peat the $p$ value of the regression is better when looking at the 20-30 cm average soil temperature ($p = 0.075$, $r^2 = 0.38$), than with the 30-40 cm temperature ($p = 0.13$, $r^2 = 0.30$). It is understandable that estimating $Q_{10}$ gets more difficult with a shallow peat column depth, because the parameter has less effect as in the summer a majority of the whole peat column is dry. For the simulations with 150 or 200 cm of peat the regression for $Q_{10}$ does not give meaningful results and the $p$ values are large, implying that a deep active peat column might pose an additional degree of difficulty for performing the $CH_4$ emissions.

The best experiments in terms of the predictability of the hierarchical variables are those with 70 and 100 cm of peat, which also were the best performing peat column depths among the model calibration results (Table 3). For those depths, the hierarchical parameters $Q_{10}$ and $\zeta_{\mathrm{exu}}$ for each year can be estimated with

$$Q_{10}^{7l} = 3.27\,\overline{T_{30-40}^{70}} + 1.71 \tag{12}$$

$$\zeta_{\mathrm{exu}}^{7l} = -56000\,NPP^{130} + 0.468 \tag{13}$$

$$Q_{10}^{7l} = 2.60\,\overline{T_{30-40}^{70}} + 2.67 \tag{14}$$

$$\zeta_{\mathrm{exu}}^{10l} = -68300\,NPP^{130} + 0.480, \tag{15}$$

where the upper indexes 7l and 10l refer to the number of 10 cm peat layers, temperatures are in °C, and the units of NPP are mol m$^{-2}$ s$^{-1}$.

The lower $p$ and higher $r^2$ values for the 70 and 100 cm models suggest that also in terms of predictive skill these configurations are superior as the hierarchically varying variables can be more robustly estimated. A leave one out-cross validation (LOO-CV, see e.g. Gelman et al. (2013)) of the predicted fluxes was therefore performed on the 70 cm and 100 cm models by optimizing them with respect to the costfunction in Eq. 11 leaving one year at a time out, calculating the estimates for the hierarchical parameters based on the results obtained, and predicting the $CH_4$ emissions for the year that was left out. The algorithm (BOBYQA) and the number of iterations completed (350) were the same as before the MCMC and for the hierarchical parameters the priors were defined by the values defining the hyperprior. The results of the cross validation are shown in Fig. 5 (g) and 6 (b). Compared to the NHPM estimate and the default parameter values the annual errors were reduced and the mean annual errors were -4.95% and -3.01% with the standard deviations of 13.6% and 13.4 %, for the 70 cm and 100 cm peat column depths respectively. In comparison to the NHPM estimate the annual errors were reduced for six out of the nine years. These results are promising, and as the analysis performed was extremely simple, there is room for further development.



# 6 Conclusions

In this study, Bayesian calibration of a new process-based wetland $CH_4$ emission model, sqHIMMELI, was performed using MCMC methods against observations from the Siikaneva flux measurement site in Finland. The results show that the modeled processes and the estimated parameters are identifiable with the flux data, with the exception of $CH_4$ oxidation. The depth of

the modeled peat column, for which an optimal range was found at around 70 cm, strongly affects the posterior distributions of the parameters, and the optimal model functioning in terms of the gas transport pathway fractions and the relative importances of the sources of the anaerobic respiration. Such dependence on the chosen model setup suggests that using strict measurement-based priors can cause the model to behave in an unintended manner. Preliminary results obtained also suggest that estimation of the annually varying $CH_4$ production-related parameters is feasible and may help to improve the future estimates of the

boreal wetland $CH_4$ emissions.

Since the flux observations were not enough to constrain the oxidation rates, a possible path forward is to estimate model and parameter states assimilating column concentrations with a method such as the one presented in Hakkarainen et al. (2012). Along that path, combining observations from several sites and optimizing them together with the methods presented here in conjunction with independent validation can provide valuable information about the uncertainties related to wetland emission

modeling and about how to best improve the quality of predicting wetland methane emissions in land surface schemes of climate models.

# 7 Code availability

The HIMMELI source code is available as a supplement to the publication Raivonen et al. (2017).

# 8 Data availability

The model input data and the flux measurement data are available upon request from the lead author.

# Appendix A: The sqHIMMELI model equations

The version of HIMMELI presented here describes processes for $CH_4$ production and transport. It differs from the version presented in Raivonen et al. (2017) in that the model presented there does not contain the processes for anaerobic respiration but rather take it as input, the idea being that such input would be available when using HIMMELI as a part of a larger model.

Hence the equations presented in Sec. A2 are specific to the version used in this study. The other difference between the models is the difference between the root distributions described in Sec. 3.2.

## A1 Governing equations

The gas concentrations of $CH_4$, carbon dioxide and oxygen in the peat column are governed by the equations





$$T_X(t,z) = Q_X^{\mathrm{diff}} + Q_X^{\mathrm{plant}} + Q_X^{\mathrm{ebu}} \tag{A1}$$

$$\frac{\partial [\mathrm{CH_4}]}{\partial t}(t,z) = -T_{CH_4} + R_{CH_4}^{\mathrm{exu}} + R_{CH_4}^{\mathrm{peat}} - R_{CH_4}^{\mathrm{oxid}} \tag{A2}$$

$$\frac{\partial [\mathrm{O_2}]}{\partial t}(t,z) = -T_{O_2} - R_{\mathrm{aerob}}^{\mathrm{peat}} - R_{CO_2}^{\mathrm{exu}} - 2R_{CH_4}^{\mathrm{oxid}} \tag{A3}$$

$$\frac{\partial [\mathrm{CO_2}]}{\partial t}(t,z) = -T_{CO_2} + R_{CO_2}^{\mathrm{exu}} + R_{CO_2}^{\mathrm{peat}} + R_{CH_4}^{\mathrm{oxid}} + R_{\mathrm{aerob}}^{\mathrm{peat}}, \tag{A4}$$

where $T_X(t,z)$ describes transport of gas $X$ containing the diffusion, ebullition, and plant transport components, and $R$ stands for production or consumption. The different terms in the equations are described below.

**A2    Anaerobic respiration producing CH$_4$**

The equations presented in this section are specific to the version of HIMMELI used in this study. The version in Raivonen et al. (2017) takes the rate of anaerobic decomposition of carbon as input and does not treat the different sources of that carbon

separately.

The carbon for methane production in this model version comes from two sources: root exudates and anaerobic peat composition. The methane production from anaerobic respiration of that carbon is given by the terms $R_{CH_4}^{\mathrm{exu}}$ and $R_{CH_4}^{\mathrm{peat}}$ described by:

$$R_{CH_4}^{\mathrm{exu}}(z) = \frac{f_{CH_4}^{\mathrm{exu}}}{\mathrm{d}z} \nu \frac{\pi(z)}{1 + \eta C_{O_2}(z)} \tag{A5}$$

$$R_{CH_4}^{\mathrm{peat}}(z) = k_{\mathrm{cato}}(z) g_{CH_4}^{Q_{10}} \frac{\rho_{\mathrm{cato}} f_{C_{\mathrm{cato}}}}{M_C}, \tag{A6}$$

where in Eq. A5 $\nu$ is the decay rate of root exudates from Eq. 3, $\eta$ is an oxygen inhibition parameter, $C_{O_2}(z)$ is the oxygen concentration at depth $z$, and $\pi(z)$ is the normalized proportion of the total anaerobic root mass, also at depth $z$, given in an unnormalized form in Eq. 2. The parameter $f_{CH_4}^{\mathrm{exu}}$ is a constant determining what fraction of root exudates in anaerobic conditions will turn into CH$_4$. Equation A5 is only used below the water table. In Eq. A6, $g_m^{Q_{10}}$ is the proportion of the

anaerobic peat decomposition process producing CH$_4$, $\rho_{\mathrm{cato}}$ is the peat density in the catotelm, $f_{C_{\mathrm{cato}}}$ is the fraction of carbon in catotelm peat, and $M_C$ is mass of carbon. The parameter $k_{\mathrm{cato}} = Q_{10}^{\frac{(T - 273.15)}{10}} / \tau_{\mathrm{cato}}$ is described in Eq. 5, and is zero above water table.

The equations for CO$_2$ are similar:

$$R_{CO_2}^{\mathrm{exu}}(z) = \nu\pi(z) - R_{CH_4}^{\mathrm{exu}}(z) \tag{A7}$$

$$R_{CO_2}^{\mathrm{peat}}(z) = (1 - g_{CH_4}^{Q_{10}}) k_{\mathrm{cato}}(z) \frac{\rho_{\mathrm{cato}} f_{C_{\mathrm{cato}}}}{M_C}, \tag{A8}$$

and the meanings of the symbols are analogous to the ones in equations for CH$_4$.





## A3 Peat respiration and methane oxidation

Peat respiration (aerobic respiration) is described with an equation of the Michaelis-Menten form

$$R_{\text{aerob}}^{\text{peat}}(z) = V_R(z)\frac{\alpha C_{O_2}^x(z)}{K_R + C_{O_2}^x(z)}, \tag{A9}$$

where $\alpha$ is a dimensionless Henry solubility constant for oxygen above the water table, and one below it, see Tang et al. (2010).
The factor $C_{O_2}^x$ refers to $C_{O_2}^w$ below the water table, and to $C_{O_2}^a$ above it. Here $w$ and $a$ refer to whether the concentration is in the gaseous or in the liquid phase. Parameter $K_R$ is the Michaelis-Menten constant of the process, and $V_R(z)$ is given by Eq. 6.

Methane oxidation is controlled by dual-substrate Michaelis-Menten kinetics,

$$R_{CH_4}^{\text{oxid}}(z) = V_O(z)\frac{C_{O_2}^x(z)}{K_{O_2} + C_{O_2}^x(z)}\frac{C_{CH_4}^x(z)}{K_{CH_4} + C_{CH_4}^x(z)}, \tag{A10}$$

and here $\alpha$ factors similar to the one in Eq. A9 have been absorbed into the concentration terms - otherwise the terms are analogous to those in Eq. A9, except for that the term $V_O(z)$ is described by Eq. 7.

## A4 CH$_4$ transport

The transport term $T_X(t, z)$ in Eq. A1 consist of the following terms:

$$Q_X^{\text{diff}} = D_{\text{medium}}^X\frac{\partial}{\partial z}C_X^{\text{medium}} \tag{A11}$$

$$Q_X^{\text{plant}}(z) = \frac{\rho\pi(z)D_{\text{air}}^X}{\tau^2}\frac{LAI}{SLA}\frac{C_x(t,z) - C_X^{\text{atm}}}{z} \tag{A12}$$

$$Q_X^{\text{ebu}}(z) = -k\sigma\frac{pp_{i,X}}{RT}\frac{\sum_i pp_i(z) - (P_{\text{atm}} + P_{\text{hyd}}(z))}{\sum_i pp_i(z)}. \tag{A13}$$

The first of these is the diffusion, where the diffusion coefficients $D$ are given by Eq. 9 and 10, and "medium" refers to either air or water. Due to coding mistake, the $f_{D,a}$ and $f_{D,w}$ coefficients in the aforementioned equations were set to 0.1 for gases other than CH$_4$ in this work.

The second equation is for plant transport, with $\rho$ and $\tau$ described in context of Eq. 8, $\pi(z)$ is the normalized root distribution mentioned above, and $C_X^{\text{atm}}$ refers to the atmospheric partial pressure of gas $X$. LAI stands for the leaf area index, given as input, and SLA is the specific leaf area. The note above regarding the $f_{D,a}$ values is also valid for plant transport, as it is a factor determining $D_{\text{air}}^X$.

The third equation is the ebullition component of the gas transport, where $pp_i$ refers to the partial pressure of different gases
indexed with $i$, $R$ is the universal gas constant, $k$ is an ebullition rate constant, and $\sigma$ is the peat porosity. The parameters $P_{\text{atm}}$ and $P_{\text{hyd}}(z)$ refer to the atmospheric pressure and hydrostatic pressure at depth $z$, respectively.

Table 5 shows the parameters that are used in the equations above but not optimized in this work, along with their values.



**Appendix B: Error model for residuals**

In section 4.2.1 we described the error model as an AR(1) model where the residuals are Laplace-distributed. The error of each measurement was described as a fraction $\alpha$ on the absolute value of the observation at that time, plus a constant error component, $\gamma$.

Let $\boldsymbol{y}' = \max(\boldsymbol{c}, \boldsymbol{y})$, where $\boldsymbol{c}$ is the 14-day running mean of the gap-filled $CH_4$ flux observations $\boldsymbol{y}$, and where the maximum is understood to be taken over each pair separately. Let $M$ be the model, $\boldsymbol{x}_t$ the model state, and $\boldsymbol{z}_t$ the forcing input data at time $t$. Then the error-scaled residual at time $t$ is

$$r_t = \frac{y_t - M(\boldsymbol{x}_{t-1}, \boldsymbol{z}_t, \boldsymbol{\theta})}{\delta_t}, \tag{B1}$$

where $\delta_t = \alpha |y_t'| + \gamma$. Let $\phi$ denote the lag-1 autocorrelation coefficient, meaning the correlation of the residual timeseries with
the same residual timeseries one day later. The AR(1)-corrected residual for time $t$ then becomes

$$r_t^* = r_t - \phi \, r_{t-1}^*. \tag{B2}$$

The motivation for the way of constructing $\boldsymbol{y}'$ above is to allow for a reasonable amount of error both in the case when there is an emission spike upwards and when the same happens downwards, avoiding the problems where in the summer there is suddenly a day with zero $CH_4$ emissions, and the likelihood would take the observation to be extremely precise (as $\alpha y_t$ would
be small) because of the low absolute value.

The model was fitted against the data with a crude least-squares likelihood in order to determine $\alpha$ and $\gamma$. The AR(1)-transformed residuals obtained follow the Laplace distribution with mean 0 and scale 1, when $\alpha = 0.08$, and $\gamma = 0.00025 \ \mu$ mol s$^{-1}$ m$^{-2}$. The AR(1) parameter $\phi$ was set to 0.2. The parameter $\delta_t$ in Eq. B1 and the annual flux term in Eq. 11 were finally scaled so that the contribution from the residuals was 80%, and the contributions from the annual fluxes and prior were
10% each in terms of the number of summands. This was done to prevent being overconfident with the parameter estimates and to account for that the value of 0.2 for $\phi$ was on the low side, since the transformed residuals still are autocorrelated with lag-1 correlation coefficient between 0.45 and 0.65 for the PM and MAP estimates. Other uncertainties motivating such treatment are uncertainties in the NPP model, inadequacies in the model description of the peat column and lack of spatial heterogeneity in the model description, filled gaps in the water table depth data, errors from interpolation of the soil temperature data and
heat transfer, and other unknown error sources. The same model error description was used for all simulations.

The residual histograms of the 70 cm PM estimates show that the error model transformed unscaled residuals closely follow the Laplace distribution. There is a slight extra concentration of mass on the lower side, which is explainable with the overall negative bias of a few percents in the emission estimate, which was shown in Fig. 6.





## Appendix C:  A basic outline of MCMC

Markov Chain Monte Carlo (MCMC) methods are a class of Bayesian methods that can be used for obtaining the probability distribution $p(\boldsymbol{\theta}|\boldsymbol{y})$ for a parameter vector $\boldsymbol{\theta} \in \boldsymbol{R}^n$ given data $\boldsymbol{y} \in \boldsymbol{R}^k$. According to Bayes' theorem from 1763, this can be written as

$$p(\boldsymbol{\theta}|\boldsymbol{y}) = \frac{p(\boldsymbol{y}|\boldsymbol{\theta})p(\boldsymbol{\theta})}{p(\boldsymbol{y})}, \tag{C1}$$

where $p(\boldsymbol{y}|\boldsymbol{\theta})$ is the *likelihood* (in this work the first two terms on the right hand side of Eq. 4.2.4), and $p(\boldsymbol{y})$ is the *prior* (the last term). The *evidence*, $p(\boldsymbol{y})$ is often very difficult to evaluate, but in MCMC this is not needed, because MCMC algorithms evaluate ratios of successive evaluations of $p(\boldsymbol{\theta}|\boldsymbol{y})$, making the denominators to cancel out and hence the evidence term can be dropped.

MCMC sampling starts by taking some starting value $\boldsymbol{\theta}$, and calculating the objective function (also known as *costfunction*) value $J(\boldsymbol{\theta}) \in \boldsymbol{R}$. The algorithm then draws a new sample of the parameter vector, $\boldsymbol{\theta}'$ from a prescribed *proposal distribution* $q(\boldsymbol{\theta})$, and evaluates $J(\boldsymbol{\theta}')$. It accepts the new parameter vector with a probability that depends on the value of $J(\boldsymbol{\theta}')$ and the objective function value of the previous accepted parameter, $J(\boldsymbol{\theta})$. If the value is accepted, the chain will move to position $\boldsymbol{\theta}'$ (setting $\boldsymbol{\theta} \leftarrow \boldsymbol{\theta}'$), and if $\boldsymbol{\theta}'$ is rejected, the value $\boldsymbol{\theta}$ will be repeated in the chain. After this a new value, sampled from $q(\boldsymbol{\theta})$ (which is possibly a different distribution from the one used at the previous iteration as $\boldsymbol{\theta}$ may have changed) will be proposed and the whole process is repeated. In the end the procedure will produce a chain of parameter values.

This chain of parameter vectors will theoretically, given infinitely many iterations, converge to the the *target distribution* of the Markov process, meaning, that in such a case picking a random element from the chain amounts to drawing a sample directly from the target distribution. As real-life Markov chains are of finite length, the *posterior distribution* obtained from the chain is an approximation of the underlying target distribution.

In practice this means, that with MCMC it is possible to find a good approximation of the probability density function of the parameter vector $\boldsymbol{\theta}$ in cases, where the model is too complicated for analytical treatment. From this probability density function, valuable information such as modes, variances, and correlations of the parameters can be analyzed. The posterior also reveals, what parameters are constrained by the data, and what are not.

For efficient convergence of the chain to the posterior distribution a good estimate of $q(\boldsymbol{\theta})$ is needed. The Adaptive Metropolis algorithm automatically calibrates the proposal during the MCMC.

## Appendix D:  Metropolis within Gibbs sampling of the parameters

The hierarchical parameters $Q_{10}^{\text{year}}$ and $\zeta_{\text{exu}}^{\text{year}}$ are denoted here generically by $\theta^i$, where $i$ refers to the different years. The priors of these parameters are defined by the *hyperparameters* $\mu_i$ and $\sigma_i$ that determine the prior of $\theta^i$ by

$$\theta^i \sim N(\mu_i, \sigma_i^2). \tag{D1}$$





The unknown hyperparameters $\mu_i$ and $\sigma_i^2$ have probabilistic models

$$\mu_i \sim N(\mu_0, \tau_0^2) \tag{D2}$$

$$\sigma_i^2 \sim \text{Inv-}\chi^2(n_0, \sigma_0^2), \tag{D3}$$

where $\mu_0$ and $\tau_0^2$ define the mean and variance of the hyperprior of $\mu_i$, $n_0 \in \mathbf{N}$ defines the number of degrees of freedom of

the Inv-$\chi^2$ distribution, and $\sigma_0^2$ is the expected value of the scaled Inv-$\chi^2$ distribution.

In Gibbs sampling the full conditional posterior distributions of the hyperparameters and the parameters $\theta_i$ are sampled in turns. Due to the conjugacy of the normal distribution and the scaled Inv-$\chi^2$ distribution, closed form expressions exists for sampling from $p(\mu_i | \sigma^2, \mu_0, \sigma_0^2, \theta^i)$ and $p(\sigma_i^2 | \sigma_0^2, n_0, \theta^i)$, where $\mu$ is the current mean of the parameters $\theta_i$ and $\sigma^2$ is their variance. The Gibbs sampling therefore consists of three steps:

1. Draw $\mu_i$ from

$$\mu_i | \mu, \sigma^2 \sim N\left( \frac{\frac{n_i \overline{\theta^i}}{\sigma^2} + \frac{\mu_0}{\tau_0^2}}{\frac{n_i}{\sigma^2} + \frac{1}{\tau_0^2}}, \frac{1}{\frac{n_i}{\sigma^2} + \frac{1}{\tau_0^2}} \right), \tag{D4}$$

    2. draw $\sigma_i^2$ from

$$\sigma_i^2 | \boldsymbol{\theta}, \boldsymbol{\mu} \sim \text{Inv-}\chi^2\left( n_0 + n_i, \frac{\sigma_0^2 n_0 + \sum_{j=1}^{n_i} (\theta_j^i - \mu_j)^2}{n_0 + n_i} \right), \text{ and} \tag{D5}$$

    3. draw the parameters $\theta_i$ (and the non-hierarchical parameters) with MCMC, since closed-form expression for $p(\boldsymbol{\theta}|\phi, \boldsymbol{y})$,

15        where $\phi$ denotes all the different hyperparameters, is not available.

In this work, the value of the parameter $\tau_0^2$ was set to the value of $\sigma_0^2$, $n_i$ is the number of years, and the value of $n_0$ was set to 9. The means and variances obtained this way describe the interannual variability of the parameters, and not including them as parameters in the MCMC sampling reduces the dimension of space that the MCMC sampler needs to explore, speeding up convergence of the posterior distribution.

**Appendix E: NPP and LAI**

We estimated the net photosynthesis rate, $P_n$, of vascular plants of Siikaneva for years 2005-2014 by utilizing regression models of gross photosynthesis, $P_g$, and autotrophic respiration $R_a$ formulated for peatland vegetation (Riutta et al., 2007a, b; Raivonen et al., 2015). The model of the $P_g$ of sedge and dwarf shrub canopy (Riutta et al., 2007a) simulates the carbon uptake driven by photosynthetically active radiation (PAR), WTD and air temperature. The model of $R_a$ (Raivonen et al., 2015)

simulates the respiration rate driven by air temperature and WTD and was parameterized for sedges only.

Both $P_g$ and $R_a$ models simulate the carbon fluxes per soil surface area and the rate depends on the LAI. We simulated the LAI using a lognormal function presented by (Wilson et al., 2007). Parameter values of the LAI model were obtained



by averaging the values reported by (Wilson et al., 2007) for the vascular species abundant at Siikaneva. For the growing season peak LAI we used the maximum LAI observed at the eddy covariance footprint area, viz. approximately 0.4 m$^2$ m$^{-2}$ (Riutta et al., 2007b). We also included a constant wintertime LAI since a significant green sedge biomass may overwinter, approximately 15% of the maximum (Saarinen, 1998; Bernard and Hankinson, 1979). The overwintering LAI at Siikaneva would thus be 0.05 m$^2$ m$^{-2}$ . The same LAI was used for all the years and this LAI also was given as the input for the CH$_4$ transport model.

The daily averages of $P_n$ were calculated by subtracting $R_a$ from $P_g$. The models were run with measured meteorological data. We determined the photosynthetically active seasons based on snowmelt dates in spring or arrival of snowcover in autumn from the reflected PAR data, or based on air temperature (permenently greater than 5 °C assumed to be the growing season). After the calculation, we compared the resulting $P_n$ of vascular vegetation of year 2005 to eddy covariance CO$_2$ fluxes from Siikaneva. We used the GPP derived from the measured NEE by (Aurela et al., 2007). This was the only available year of processed CO$_2$ flux data. The GPP was on average 4.5-fold compared with our $P_n$ , with a R$^2$ of 0.9. GPP also includes the photosynthesis of *Sphagnum* mosses as well as CO$_2$ released in autotrophic respiration. *Sphagnum* accounted for 20-40% of the GPP in the study by (Riutta et al., 2007a) and autotrophic respiration has been observed to be roughly 50% of GPP (Gifford, 1994). Consequently, the NPP of vascular vegetation can be estimated by multiplying the GPP with $0.7 \times 0.5$. This estimate was still 1.56-fold compared with the $P_n$ for the year 2005. Since the $P_n$ also was lower than generally reported for peatlands, we chose to trust the eddy covariance measurement and scaled the $P_n$ of all the years upwards by multiplying with 1.56. For further details, please consult Raivonen et al. (2017).

## Appendix F: Supplementary information

### F1    Details regarding the AM algorithm usage

In order to infer about the posterior distribution, the MCMC chain needs to be long enough, and converged to produce the right statistics. The MCMC chains driven by the AM algorithm mixed well, example of which as can be seen in Fig. 12 showing the chain from the experiment with 100 cm of peat. The proposal distribution of the AM algorithm was adapted when the iteration number was a square of an integer, and for the adaptation 20% from the start of the chain was discarded. In the early stages of each experiment, the initial approximation for the proposal covariance, calculated from the Jacobians of the model, was allowed to dominate until after accepting enough proposed points there was sufficient data to start the proposal covariance adaptation procedure.

### F2    Computational requirements

Even though the model runs fast, in around five to thirty seconds for the ten-year period on a multicore laptop, due to the large number of simulations, the MCMC experiments needed to be performed on a CRAY XC-20 supercomputer using a single node for a single MCMC chain and running all the experiment in the RAM of the computer minimizing hard drive utilization.





Shared-memory parallellization was used to run the different years at the same time and the MCMC experiments were run for a month during which all the MCMC experiments completed between 78000 and 391000 forward model simulations.

*Author contributions.* Jouni Susiluoto designed the study with help from the co-authors, programmed the algorithms, performed the model simulations, analyzed the results, and prepared the manuscript and the figures. Maarit Raivonen provided and validated the input data and

5  helped with the interpretation of the results. Leif Backman contributed several model subroutines and helped to interpret the results. Marko Laine provided assistance with getting the technical aspects of the Bayesian analysis right. Olli Peltola provided insight into the data used. Jarmo Mäkelä, Timo Vesala, and Tuula Aalto provided helpful critical comments and suggestions that helped to improve the manuscript substantially.

*Competing interests.* The authors state that they are free from any conflicting interests.

10  *Acknowledgements.* We would like to thank the University of Helsinki researchers M.Sc. Pavel Alekseychik, and Dr. Ivan Mammarella, and Prof. Janne Rinne from Lund University for valuable comments regarding the Siikaneva measurement site data. We would also like to thank Prof. Heikki Haario from Lappeenranta University of Technology, Dr. Janne Hakkarainen from Finnish Meteorological Institute, and Prof. Samuli Siltanen from University of Helsinki for comments regarding the mathematical aspects of the study.

This work has been supported by the EU LIFE+ project MONIMET LIFE12 ENV/FI/000409, and the EU FP7 project EMBRACE. We

15  additionally acknowledge funding from the RED platform of the Lappeenranta University of Technology, and thank the Academy of Finland Center of Excellence (272041), CARB-ARC (285630), ICOS Finland (281255) ICOS-ERIC (281250), NCoE eSTICC (57001), EU-H2020 CRESCENDO (641816) and Academy Professor projects (284701) and (282842).



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



**Table 1.** Description of the data used

| Data | Description | Usage | Units | Source | Comments |
|------|-------------|-------|-------|--------|----------|
| LAI | leaf area index | input | - | modeled | Gaussian curve to approximate the seasonal cycle |
| WTD | water table depth | input | m | measured | gap-filled at various times |
| NPP | net primary prod. | input | $\mathrm{mol\ m^{-2}\ s^{-1}}$ | modeled | generated by a separate NPP model |
| $T_{\mathrm{soil}}$ | soil temperature | input | °C | measured | interpolated from fewer observation depths |
| $CH_4$ | $CH_4$ flux | objective function | $\mathrm{mol\ m^{-2}\ s^{-1}}$ | measured | used in the objective function formulation |



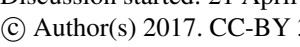

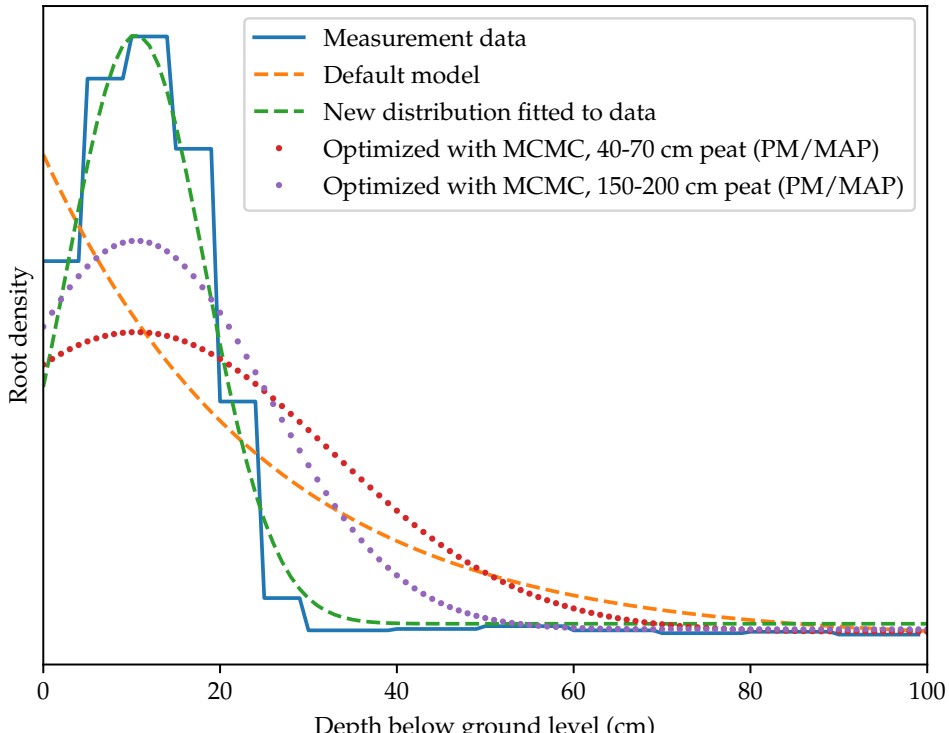

**Figure 1.** The different root distribution descriptions. The original description is shown as the decaying exponential, and the graph with discrete steps shows measurement data from Saarinen (1996). The new root distribution curve with optimized parameters are shown along with the curves resulting from the MCMC optimization. The original distribution gives more root mass to depths of 50-80cm, than the MCMC-optimized curves of the new root distribution. All curves are normalized to the same total root mass.





| | Low | High | Units | Prior $\mu$ | Prior $\sigma$ | Source |
|---|---|---|---|---|---|---|
| $f_{D,a}$ | 0.01 | 1.0 | - | 0.1 | 0.2 | (Iiyama and Hasegawa, 2005) |
| $f_{D,w}$ | 0.1 | 1.0 | - | (0.5) | - | - |
| $V_{R0}$ | $2 \times 10^{-5}$ | $3 \times 10^{-4}$ | mol m$^{-3}$ s$^{-1}$ | $4 \times 10^{-5}$ | $2 \times 10^{-5}$ | (Nedwell and Watson, 1995; Watson et al., 1997) |
| $V_{O0}$ | $2 \times 10^{-5}$ | $3 \times 10^{-4}$ | mol m$^{-3}$ s$^{-1}$ | $1 \times 10^{-4}$ | $6 \times 10^{-4}$ | Order of magnitude from (Segers, 1998), almost flat prior. |
| $\lambda_{\mathrm{root}}$ | 0.01 | 0.4 | m | 0.125 | 0.05 | Fitted to data in (Saarinen, 1996) |
| $\tau$ | 1.0 | 5.0 | m m$^{-1}$ | 1.5 | 0.5 | (Stephen et al., 1998) |
| $\rho$ | 0.05 | 0.4 | m$^2$ kg$^{-1}$ | 0.085 | 0.0425 | (Stephen et al., 1998) |
| $\tau_{\mathrm{exu}}$ | 3 | 30 | days | 14 | 2.5 | (Wania, 2007) |
| $\tau_{\mathrm{cato}}$ | 1000 | 30000 | years | (11111) | - | Value affected by peat depth, hence using a flat prior. |
| $Q_{10}$ | 1.7 | 16.0 | - | 5.9 | 2.0 | (Juottonen, 2008; Gedney et al., 2004; Bergman et al., 2000) |
| $\zeta_{\mathrm{exu}}$ | 0.01 | 0.99 | - | 0.2 | 0.2 | Lower bound of (Walker et al., 2003) |

**Table 2.** Parameter limits and prior distribution parameters. The priors are truncated Gaussian, with mean values $\mu$ and standard deviations $\sigma$, truncated at the values in the columns *low* and *high*. The values in the "Prior $\mu$" column are used as values in the "default" simulations, results from which are shown in e.g. Fig. 8 and 5.



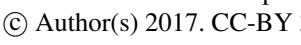

**Figure 2.** Posterior distributions of the parameters from the MCMC. The two-dimensional marginal distributions of the posterior distribution from the experiment with 70 cm peat depth are shown in the triangle on the lower left (labels on the left and at the bottom), and the correlations between parameters are shown in the upper triangle on the right (labels on the left and on the top). The images in the lower left triangle show the 90% (black) 50% (red), and 10% (blue) contours, and roughly a one-thousandth of the points sampled. The first 50% of each chain was discarded as a warm-up period. On the upper right, each plot shows correlation coefficients between parameters for all experiments, color-coded to show negative correlations in blue and positive in red. Left-to-right and top to bottom in each color-coded square, the depths are 40, 50, 70, 100, 150, and 200 cm of peat. The values from the posterior of the 7-layer experiment, referring to 70 cm peat and shown also in the lower left part of the figure, are marked in boldface.





**Table 3.** Parameter values obtained in the optimization of the models with the different peat depths. Values shown are for the maximum a posteriori (MAP) / posterior mean (PM) estimates. The horizontal line in the middle separates the hierarchically optimized parameters (including their priors) from the others.

| Peat depth | 40 cm | 50 cm | 70 cm | 100 cm | 150 cm | 200 cm |
|---|---|---|---|---|---|---|
| $\tau_{\mathrm{cato}} \times 10^{-3}$ (y) | 2.35 / 3.02 | 3.02 / 4.00 | 5.10 / 6.04 | 9.12 / 10.1 | 15.9 / 15.1 | 21.9 / 24.7 |
| $\tau$ | 1.29 / 1.31 | 1.13 / 1.28 | 1.18 / 1.19 | 1.05 / 1.14 | 1.01 / 1.05 | 1.04 / 1.07 |
| $\tau_{\mathrm{exu}} \times 10^{-5}$ (s) | 11.6 / 10.8 | 9.86 / 9.73 | 8.67 / 8.83 | 9.47 / 8.75 | 8.56 / 8.98 | 7.05 / 7.15 |
| $f_{D,w}$ | 0.78 / 0.55 | 0.55 / 0.56 | 0.65 / 0.64 | 0.74 / 0.81 | 0.99 / 0.80 | 0.60 / 0.74 |
| $f_{D,a}$ | 0.58 / 0.50 | 0.55 / 0.54 | 0.52 / 0.54 | 0.72 / 0.65 | 0.86 / 0.84 | 0.73 / 0.84 |
| $\lambda_{\mathrm{root}}$ | 0.30 / 0.30 | 0.28 / 0.29 | 0.33 / 0.32 | 0.34 / 0.35 | 0.21 / 0.21 | 0.23 / 0.20 |
| $\rho$ | 0.14 / 0.15 | 0.15 / 0.15 | 0.15 / 0.16 | 0.21 / 0.18 | 0.23 / 0.24 | 0.24 / 0.22 |
| $V_{R0} \times 10^{5}$ | 2.86 / 3.92 | 2.15 / 3.63 | 3.97 / 3.67 | 2.83 / 4.49 | 2.86 / 3.31 | 2.52 / 4.43 |
| $V_{O0} \times 10^{4}$ | 2.87 / 1.31 | 1.63 / 1.50 | 2.95 / 1.86 | 1.45 / 1.13 | 0.866 / 1.06 | 0.233 / 1.51 |
| $\overline{\zeta_{\mathrm{exu}}}$ | 0.49 / 0.38 | 0.38 / 0.31 | 0.34 / 0.28 | 0.31 / 0.26 | 0.38 / 0.33 | 0.23 / 0.30 |
| $\zeta_{\mathrm{exu}}^{\mathrm{std}}$ | 0.15 / 0.18 | 0.12 / 0.17 | 0.13 / 0.17 | 0.15 / 0.18 | 0.26 / 0.18 | 0.20 / 0.18 |
| $\overline{Q_{10}}$ | 1.85 / 3.80 | 3.59 / 4.58 | 3.81 / 4.85 | 4.22 / 5.15 | 5.22 / 4.90 | 6.16 / 6.78 |
| $Q_{10}^{\mathrm{std}}$ | 1.55 / 1.70 | 1.88 / 1.74 | 1.35 / 1.73 | 1.11 / 1.70 | 1.26 / 1.69 | 1.46 / 1.91 |
| $\zeta_{\mathrm{exu}}^{2006}$ | 0.53 / 0.34 | 0.40 / 0.28 | 0.25 / 0.24 | 0.23 / 0.17 | 0.30 / 0.28 | 0.25 / 0.26 |
| $\zeta_{\mathrm{exu}}^{2007}$ | 0.44 / 0.24 | 0.17 / 0.18 | 0.13 / 0.13 | 0.10 / 0.10 | 0.09 / 0.18 | 0.09 / 0.16 |
| $\zeta_{\mathrm{exu}}^{2008}$ | 0.48 / 0.35 | 0.40 / 0.31 | 0.37 / 0.27 | 0.28 / 0.25 | 0.33 / 0.31 | 0.19 / 0.29 |
| $\zeta_{\mathrm{exu}}^{2009}$ | 0.51 / 0.42 | 0.52 / 0.30 | 0.29 / 0.31 | 0.37 / 0.29 | 0.34 / 0.37 | 0.24 / 0.35 |
| $\zeta_{\mathrm{exu}}^{2010}$ | 0.65 / 0.50 | 0.52 / 0.37 | 0.35 / 0.35 | 0.48 / 0.33 | 0.32 / 0.39 | 0.25 / 0.36 |
| $\zeta_{\mathrm{exu}}^{2011}$ | 0.55 / 0.45 | 0.52 / 0.37 | 0.31 / 0.31 | 0.27 / 0.25 | 0.18 / 0.22 | 0.18 / 0.18 |
| $\zeta_{\mathrm{exu}}^{2012}$ | 0.50 / 0.37 | 0.42 / 0.32 | 0.39 / 0.34 | 0.32 / 0.30 | 0.33 / 0.38 | 0.29 / 0.32 |
| $\zeta_{\mathrm{exu}}^{2013}$ | 0.59 / 0.51 | 0.59 / 0.42 | 0.45 / 0.38 | 0.55 / 0.44 | 0.54 / 0.58 | 0.51 / 0.53 |
| $\zeta_{\mathrm{exu}}^{2014}$ | 0.43 / 0.39 | 0.59 / 0.30 | 0.29 / 0.30 | 0.32 / 0.26 | 0.40 / 0.42 | 0.36 / 0.37 |
| $Q_{10}^{2006}$ | 1.87 / 2.88 | 2.42 / 3.50 | 3.19 / 3.90 | 3.95 / 4.57 | 4.18 / 3.96 | 4.56 / 5.33 |
| $Q_{10}^{2007}$ | 2.33 / 3.96 | 4.02 / 4.28 | 4.11 / 4.38 | 4.67 / 4.74 | 5.25 / 4.52 | 5.80 / 6.30 |
| $Q_{10}^{2008}$ | 2.29 / 3.24 | 2.96 / 3.67 | 3.09 / 4.14 | 3.97 / 4.23 | 3.57 / 3.80 | 5.53 / 4.90 |
| $Q_{10}^{2009}$ | 2.82 / 3.77 | 2.81 / 5.02 | 4.27 / 4.89 | 4.31 / 5.02 | 4.98 / 4.74 | 6.45 / 6.53 |
| $Q_{10}^{2010}$ | 2.29 / 3.34 | 3.38 / 4.64 | 4.49 / 4.86 | 4.11 / 5.16 | 5.99 / 5.19 | 7.22 / 7.50 |
| $Q_{10}^{2011}$ | 2.58 / 3.41 | 2.90 / 4.26 | 4.37 / 4.71 | 4.92 / 5.49 | 6.73 / 6.27 | 7.65 / 9.03 |
| $Q_{10}^{2012}$ | 2.89 / 3.53 | 3.34 / 4.15 | 3.82 / 4.40 | 4.65 / 5.00 | 5.96 / 4.98 | 6.38 / 7.55 |
| $Q_{10}^{2013}$ | 4.43 / 4.82 | 3.76 / 5.72 | 5.05 / 6.30 | 4.17 / 5.72 | 5.71 / 5.23 | 6.77 / 7.63 |
| $Q_{10}^{2014}$ | 3.46 / 3.76 | 2.56 / 4.96 | 4.99 / 5.29 | 4.82 / 5.93 | 5.24 / 4.72 | 6.35 / 7.10 |
| Costfunction value | 294.85 / - | 274.37 / - | 258.93 / - | 261.58 / - | 266.83 / - | 268.67 / - |
| Model simulations | 391000 | 340000 | 277000 | 171000 | 92000 | 78000 |



**Figure 3.** Posterior marginal and prior distributions for all MCMC experiments and all parameters: (a-d) are the production-related, (e-f) the respiration and oxidation related, and (g-k) the gas transport related parameters. The curves shown for the MCMC experiments are smoothed slightly using Gaussian kernel estimates for readability. To make these figures, 50% from the start of each chain was discarded as warm-up. The dotted vertical lines show the default parameter values. For the parameters $\overline{\zeta_{\text{exu}}}$ (b) and $\overline{Q_{10}}$ (d), the prior distribution drawn is the hyperprior.



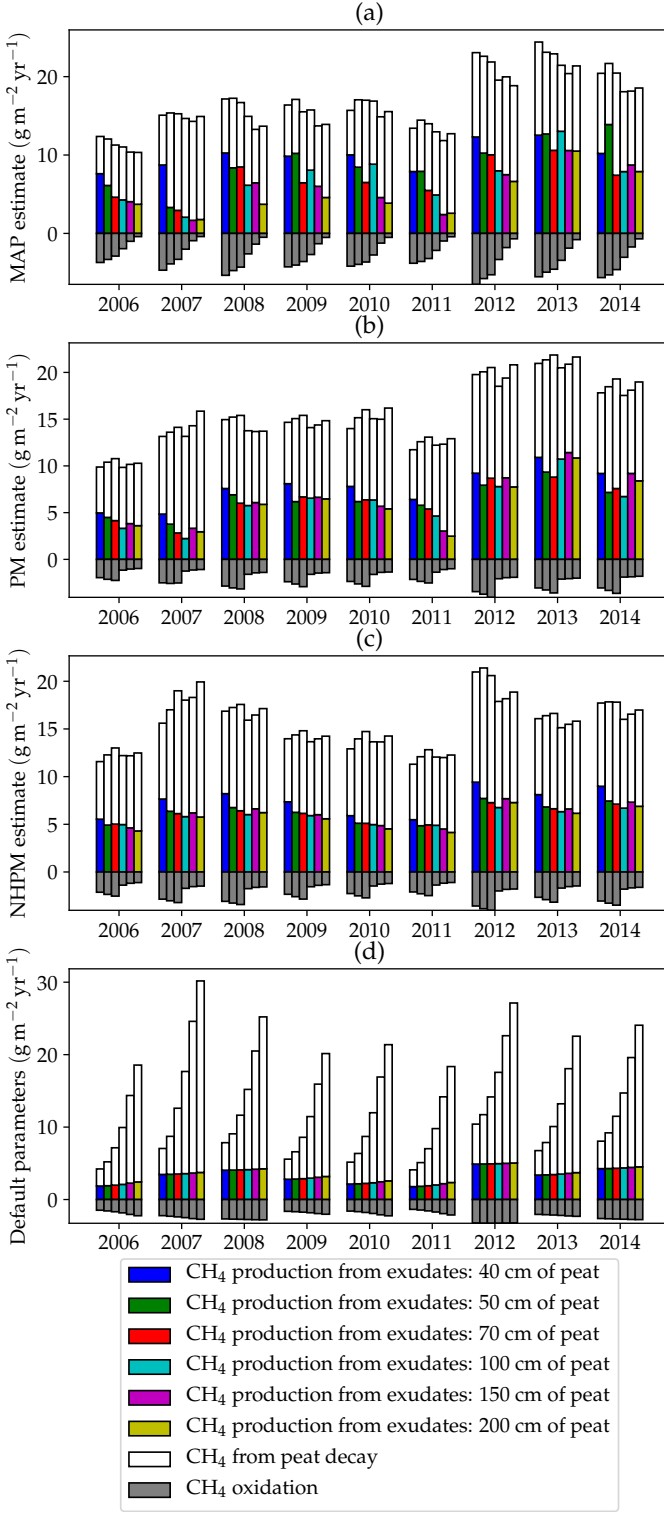

**Figure 4.** Annual $CH_4$ production in g m$^{-2}$ from root exudates (colored part) and peat decomposition (white part). Oxidized $CH_4$ is shown as gray and negative. (a) shows MAP estimate, (b) the posterior mean, (c) the non-hierarchical posterior mean estimate, and (d) the results with the default parameter values.



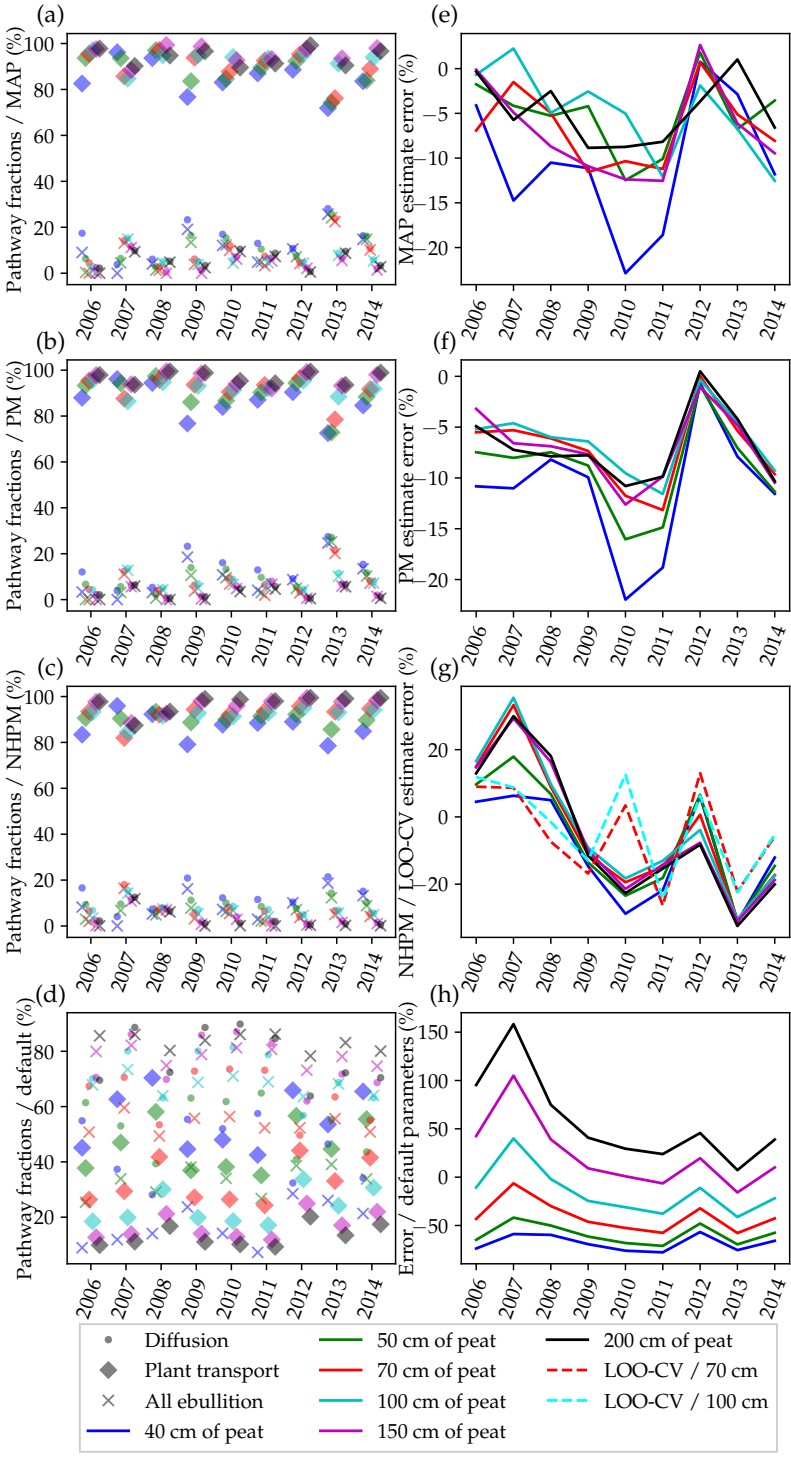

**Figure 5.** On the left side (a-d): proportions of flux components as a function of the year. Diamonds are for plant transport, dots are for the diffusion flux, and crosses describe the total ebullition taking place. The figures on the right (e-h) show the annual model-observation mismatch in percents, where only residuals from days with observation data available have been taken into account. The x-axes of sub-figures (a-d) have been spread slightly for readability. The dashed lines in sub-figure (g) represent the results from the cross validation discussed in Sec. 5.7.



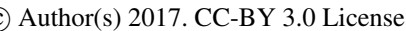

**Figure 6.** (a) Fractions of the annual diffusive fluxes of the total fluxes. Means and 1-$\sigma$ error bars are shown. Almost all ebullition takes place when the water table is below the peat surface and hence it is emitted to the atmosphere as part of the diffusion flux. Plant transport is not shown, as it is very close to the complement of the diffusive flux: together these two streams add up to more than 98% of the total flux. Plant transport variation is very close to that of diffusion. Part (b) shows the average annual errors with similar 1-$\sigma$ errorbars for the interannual variation. The results of the cross validation discussed in Sec. 5.7 are drawn in orange.





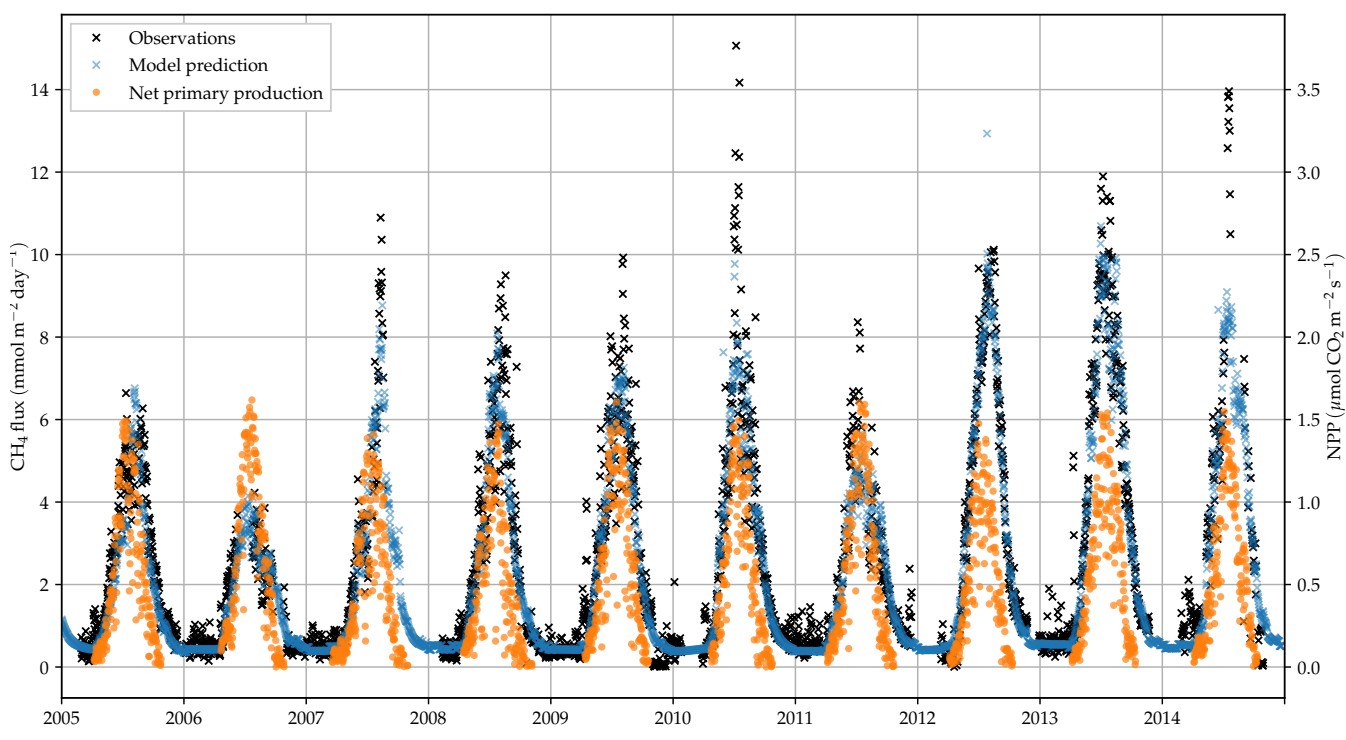

**Figure 7.** Output fluxes with parameters from the posterior mean, with 100 cm of peat. The constituents of the total flux are shown in Fig. 9.





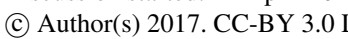

**Figure 8.** Means of total $CH_4$ emission (a), its components (b-c), total ebullition taking place (d), $CH_4$ production (e-f), $CH_4$ oxidation (g), and model residuals (h) as functions of water table depth. Shaded areas show the 5th and 95th percentiles. To look at the effect of the optimization, compare the black and the green lines.



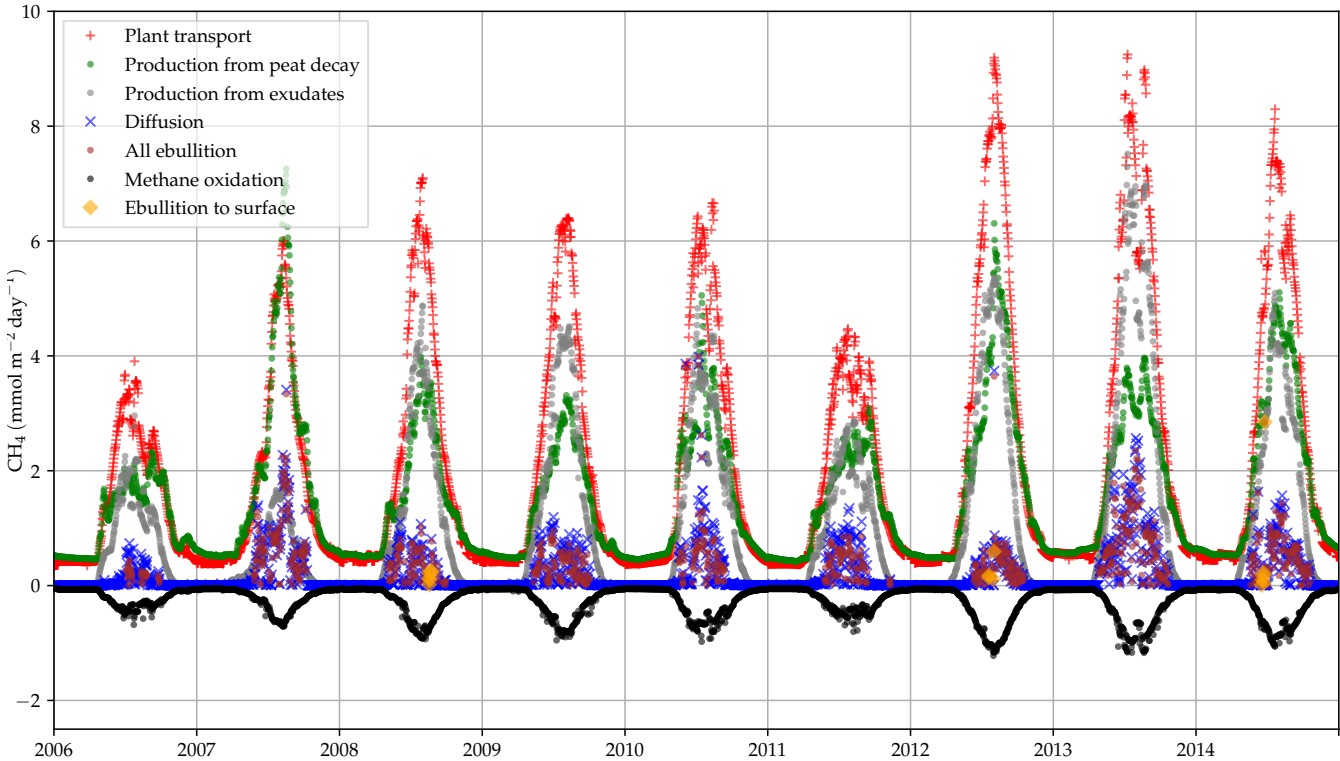

**Figure 9.** Diffusion, plant transport, ebullition, $CH_4$ production, and $CH_4$ oxidation time series for parameter values from the posterior mean estimate with 100 cm of peat. The total flux and the observations are shown in Fig. 7.



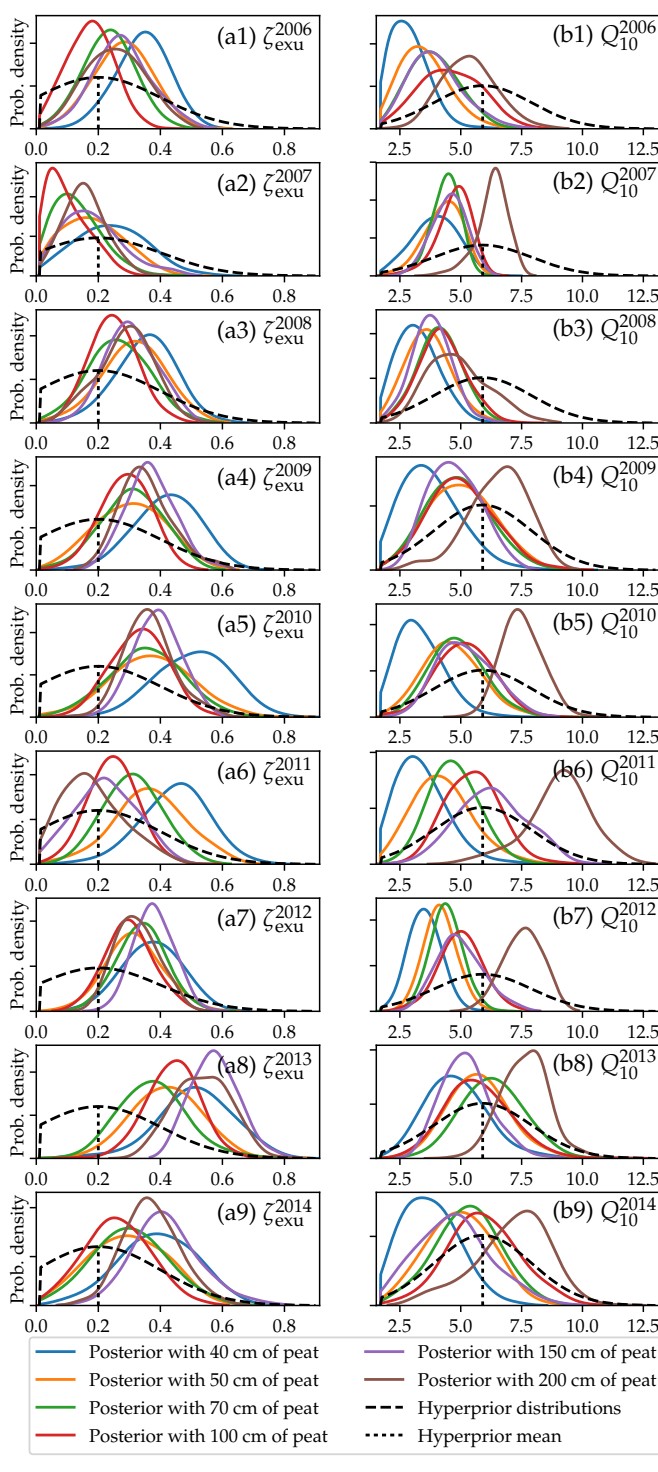

**Figure 10.** Posterior marginal distributions of the hierarchical parameters for all MCMC experiments along with the hyperpriors. The (ax) sub-figures are for the parameters $\zeta$, and the (bx) for $Q_{10}$. The curves shown for the MCMC experiments are smoothed slightly using Gaussian kernel estimates for readability. To make these figures, 50% from the start of each chain was discarded as warm-up. The dotted vertical lines show the default parameter values.



**Figure 11.** Posterior distributions and correlations of the annual means of the output from the modeled processes for the year 2008 with 70 cm peat. The dynamics for the other years and peat depths are mostly similar. The results shown are based on 1000 random samples from the parameter posterior distribution. The two-dimensional marginal distributions in the triangle on the lower left have their labels on the left and at the bottom, and the correlations between the processes in the upper triangle on the right have their labels on the left and on the top. The images in the lower left triangle show the 90% (black) 50% (red), and 10% (blue) contours.



**Table 4.** $p$ and $r^2$ values of the regressions of $Q_{10}$ against the mean soil temperature of the 10 first weeks of the year at the depth of 35 cm, and $\zeta_{\mathrm{exu}}$ against the sum of net primary production of the first 130 days of the year.

| Peat depth | $p_{Q_{10}}$ | $r^2_{Q_{10}}$ | $p_{\zeta_{\mathrm{exu}}}$ | $r^2_{\zeta_{\mathrm{exu}}}$ |
|---|---|---|---|---|
| 40 cm | 0.13 | 0.30 | 0.040 | 0.47 |
| 50 cm | 0.027 | 0.53 | 0.070 | 0.39 |
| 70 cm | 0.011 | 0.63 | 0.015 | 0.60 |
| 100 cm | 0.0068 | 0.67 | 0.027 | 0.53 |
| 150 cm | 0.34 | 0.13 | 0.040 | 0.48 |
| 200 cm | 0.17 | 0.25 | 0.043 | 0.46 |





**Table 5.** Parameters that were not part of the optimization procedure

| Parameter | Equation | Value | Units | Description | Source |
|---|---|---|---|---|---|
| $\Delta E_R$ | 6 | 50000 | J mol$^{-1}$ | heterotrophic respiration parameter | Nedwell and Watson (1995) |
| $\Delta E_{\mathrm{oxid}}$ | 7 | 50000 | J mol$^{-1}$ | CH$_4$ oxidation parameter | Nedwell and Watson (1995) |
| $f_{CH_4}^{\mathrm{exu}}$ | A5 | 0.25 | - | root exudates to CH$_4$ fraction | close to Riley et al. (2011) |
| $g_{CH_4}^{Q_{10}}$ | A6 | 0.4 | - | peat decay to CH$_4$ fraction | Schuldt et al. (2013) |
| $K_R$ | A9 | 0.22 | mol m$^{-3}$ | Michaelis-Menten coeff. | Nedwell and Watson (1995) |
| $K_{CH_4}$ | A10 | 0.44 | mol m$^{-3}$ | Michaelis-Menten coeff. | Nedwell and Watson (1995) |
| $K_{O_2}$ | A10 | 0.33 | mol m$^{-3}$ | Michaelis-Menten coeff. | Nedwell and Watson (1995) |
| SLA | A12 | 23 | m$^2$kg$^{-1}$ | specific leaf area | Vile et al. (2005) |
| $k$ | A13 | $\log(2)/1800$ | s$^{-1}$ | ebullition rate constant | - |
| $\sigma$ | A13 | 0.8 | - | peat porosity | Rezanezhad et al. (2016) |




**Figure 12.** MCMC chains from the experiment with 100 cm of peat. The hierarchical parameters in (b) and (d) show the mean value in the middle as a black mass, and the colorful surroundings are the values parameters for the individual years. The last figure (l) shows the value of the objective function.