# Peer review of "Calibrating a wetland methane emission model with hierarchical modeling and adaptive MCMC"

_Geoscientific Model Development, 2017_

## Short Comment (SC1) · 25 Apr 2017

Dear authors,

in my role as Executive editor of GMD, I would like to bring to your attention our Editorial version 1.1:

http://www.geosci-model-dev.net/8/3487/2015/gmd-8-3487-2015.html

This highlights some requirements of papers published in GMD, which is also available on the GMD website in the 'Manuscript Types' section:

http://www.geoscientific-model-development.net/submission/manuscript_types.html

[Figure]

In particular, please note that for your paper, the following requirements have not been met in the Discussions paper:

- "The main paper must give the model name and version number (or other unique identifier) in the title."

In order to simplify reference to your developments, please add a model name (sqHIM-MELI) and a version number in the title of your article in your revised submission to GMD.

Yours,

Astrid Kerkweg

––––––––––––––––––––––––––––––––––––

---

## Referee Comment (RC1) · Anonymous Referee #1 · 30 May 2017

Review of Calibrating a wetland methane emission model with hierarchical modeling and adaptive MCMC, Geoscientific Model Development

The authors use a reduced order model representation of a complex numerical wetland methane model with adaptive MCMC to estimate posterior distributions of model parameters. Because wetland methane models are complex and surface CH4 emissions are small differences between large gross fluxes, this type of calibration exercise is difficult and potentially valuable.

The authors did a nice job describing their results, given their model setup and assumptions. At this point, however, several assumptions in the approach leave me unconvinced of the reasonableness of the results, which I describe below. Also, the

underlying model has not been described in the literature, and citing "Raivonen et al. (2017)" is inappropriate, since it is apparently a paper in preparation.

Major comments 1. It is unclear why you need to vary the peat depth in your optimization. You apparently have measurements of the depth (lines 22-23, page 4), so you ought to use that as a non-calibratable value in the model. The high sensitivity of your parameter calibration to the peat depth implies to me that some other factor must be important and not properly resolved in your underlying model (e.g., $O_2$ profiles below the WT depth or rooting profiles). The argument on line 31, page 6 that it is more computationally expensive to run with a deeper peat depth is not sufficient to justify this approach. 2. Changing z_exu and Q10 on a yearly basis seems arbitrary. For Q10, I would expect much larger seasonal than inter-annual variations, yet you ignore that possibility. I think you need to explicitly describe the mechanisms you are proposing for the inter-annual variability of these parameters. The citation to Bergman et al 2000, who noticed change in Q10, "even within a single year", seems to bely your approach. 3. Ignoring the temperature sensitivity of $CH_4$ oxidation appear to be a flaw in your approach. There is substantial evidence that this temperature dependence is even larger than that for $CH_4$ production. Further, since the net surface emission is a small balance of production and oxidation rates, and you explicitly account for the $CH_4$ production temperature sensitivity (eq. (5)), not including Delta(E_R) in your equation (7) seems unreasonable. a. This concern propagates to the last sentence in your abstract. Oxidation often strongly affects the net $CH_4$ emission, which is the measurement you are using to calibrate. If parameters affecting oxidation are 'not identifiable', then it seems unlikely that the production parameters are reasonable. 4. On line 21 of page 7, you state that V_R0 affects the rate of temperature dependent HR, but the T dependency is actually governed by Delta(E_R) which is not used in the calibration (Table 2). In general, it is unclear in your section 3.4 how the $CH_4$ production occurs and its relationship with heterotrophic respiration. 5. In section 4.1, you say the model was linearized, but you did not show whether such a linearization is a reasonable approach. Please provide a quantitative evaluation of how appropriate this linearization

is. 6. Lines 22-25, page 9: Having to restart the model on January 1 of each year because realistic column gas concentrations were otherwise not predicted is a red flag for a problem in the model. The model should be able to run continuously without interruption. If this is a real problem in the model, you should rectify it. Once rectified and described in the paper, restarting each year for computational efficiency and parallelization is reasonable. 7. Section 4.2.4 seems to apply that your objective function is only based on annual values, but the text implies that you use the annual values to linearize the model, and then perform the parameter calibration with observed daily CH4 emissions. Please clarify.

Smaller comments: 1. Methane is the second most important anthropogenic GHG for warming (don't forget water vapor). 2. You describe annual calibration in they abstract, but not the fact that you used a ROM and then daily fluxes for calibration (as far as I can tell). This approach should be described in the abstract. 3. Line 13-15, page 2: cite recent methane model inter-comparisons here: Melton et al., Bohn et al. 4. Your assertion (lines 33-34, page 2 to line 4, page 3) that flawed physics representations, numerical errors, and coding errors are good reasons to calibrate a model is shocking. Calibrating a poorly constructed model is a cardinal sin of modeling, although it is regularly done. I think you might re-think the organization of this paragraph. 5. Line 5, page 3: Possibly the most mechanistic and realistic terrestrial CH4 model available today is ecosys (Grant, 2002), which you should cite. 6. Line 8, page 3: define 'multi-modality' 7. Your figures are cited out of order in the text (e.g., figure 11 cited just after figure 2). 8. Line 27, page 12: do you mean 'inter-annual variability' instead of 'annual variability'? 9. Line 9, page 12: there is no figure 6g. 10. Does the model calculate the peat temperature? It is not clear from your description which T you are using to estimate your temperature sensitivity. Air T? 11. What happened to a discussion of figure 10? 12. Where did the NPP come from? Describe in Methods.

---

## Referee Comment (RC2) · Anonymous Referee #2 · 2 Jun 2017

The objective of this paper is to use observed carbon flux time series in order to optimize parameters of a peatland carbon flux model. In general, this is a timely and important work. However, I found several serious issues with this manuscript including potential flaws in the method that does not allow a publication in the present form.

Please, indicate in abstract and introduction, what is the overall objective of this model e.g. in future applications? Do you want to apply it exclusively for this one peat site and for which question? Do you want to apply it on a continental to global scale, e.g. as part of a land surface scheme? In the latter case, several model assumptions are not useful (effective peat depth, C pool-independent decomposition flux), and a lot of work

on parameter optimization seems to be questionable when only data from one specific site is used.

Major comments to sqHIMMELI assumptions:

In both aerobic and anaerobic cases, organic matter decay seems to be a constant parameter not depending on substrate availability. When peat depth would have been set constant to the observed site-level value, then this could be valid for the specific site but then we do not learn anything from the parameter optimization procedure for a generally applicable dynamic model.

Effective peat depth: This assumption makes no sense at all. Peat depth should be a constant value corresponding to the site observation. See below for related flaws in eq 5.

A2 Anaerobic respiration producing CH4: It seems from eq. A5-A8 that you apply a CH4:CO2 ratio of 1:1 for anaerobic decomposition of root exudates. If so, please make this statement explicit and cite experimental literature showing this ratio. In section 3 it is also fully unclear if you consider anaerobic CO2 production or not.

Eq. 5: is tau(cato) the mean residence time at 273.15 K? The unit (y) in Tab 3 is not correct because in eq. 5 you do not multiply with a C pool.

I expect the Finland peat being frozen with snowpack above over long time periods of the year. What are the effects of <273K soil temperature on aerobic and anaerobic decomposition? What are the effects of soil ice on gas transport and what are the effects of snow on gas transport?

Please include in results and discussions the exudate pool values.

Peat depth: Prescribing an effective peat depth will hinder any application of that model in larger dynamic models, such as land surface schemes or DGVMs. Peat depth is no parameter there that you can prescribe but included into the mass balance equations. If you define an effective peat depth then this would mean that you either introduce a

fully recalcitrant carbon pool (case peat depth > effective peat) or that you "produce" CH4 and CO2 from non-existing carbon (case peat depth < effective peat depth). That is not a valid and also not useful model assumption.

Major comments on the parameter optimization:

Tab 5: What is the reason for not including these parameters into MCMC optimization? I generally think that the information content in the data is far too low for an optimization of all model parameters, hence a selection will be useful. However, we need good reasons for such selection, either based on theory or based on a previous sensitivity analysis.

I assume there is additional CO2 flux data available at the site. It is totally unclear why this data has not been used for constraining in addition to CH4 parameters such as decomposition and transport parameters as well as oxidation parameters parameter values cannot be transferred to other similar models and even not to HIMMELI because of the peat depth parameter and because of important differences in model formulations: root depth distribution, decomposition parameterization, etc. What is the scientific value of the paper then? Do you plan to use this model version in future studies and not the HIMMELI model?

section A4: I do not understand the sentence "Due to coding mistake, the fD,a and fD,w coefficients in the aforementioned equations were set to 0.1 for gases other than CH4 in this work." Why do you set both parameters to 0.1? With a huge pore volume in peat soils I would expect a value of 0.8 or 0.9. If that is a tuning parameter then you should optimize it. These parameters are also not listed in Tab 5. Instead they are part of Fig 2 and this seems to be a real flaw in the procedure?

For clarity, please put units on all parameters in tables and figures or when describing parameters in the text.

I cannot understand the a posteriori optimized parameter values of tau_exu in the order

of magnitude 0.00001 s (tab 3) when range is 3 to 30 days with a prior of 14 days (tab 2). From Fig 3 it seems there is a mistake in units in the table. I have similar problems with units of a posteriori V_0R which seems to be far too high. Zeta_exu seems to be with 0.5 also quite high and it would be good to see some comparison to literature values if available in the discussion.

tau(cato) ranges from 2000 to 20000 years (unit in tab 3 wrong however) depending on peat depth just because the model invalidly does not take the carbon pool into account for calculating the decomposition flux (eq. 5). Then of course, the deeper the peat the more C available the higher you need to have turnover time for the same flux. This is not a valid approach for a dynamic model.

A minor comment: I do not find it useful to have some methods description in the main text and some in appendix A but both relate so strong to each other that one understands it only when reading both together. Please move appendix A into main methods text.

---

## Author Comment (AC1) · 31 Oct 2017

Editor comments

E1: The main paper must give the model name and version number (or other unique identifier) in the title.

A: The final paper contains the model name and version number in the title.

Anonymous Referee #1

The authors use a reduced order model representation of a complex numerical wetland methane model with adaptive MCMC to estimate posterior distributions of model

parameters. Because wetland methane models are complex and surface CH4 emissions are small differences between large gross fluxes, this type of calibration exercise is difficult and potentially valuable.

The authors did a nice job describing their results, given their model setup and assumptions. At this point, however, several assumptions in the approach leave me unconvinced of the reasonableness of the results, which I describe below. Also, the underlying model has not been described in the literature, and citing "Raivonen et al. (2017)" is inappropriate, since it is apparently a paper in preparation.

A: We thank the Anonymous Referee #1 for her/his constructive critique and we address the points raised below. Here, we would like to point out that Raivonen et al. Is in public discussion in GMD, and therefore available for consultation. The list of references that we provide at the end of the article has the old reference to "in preparation" paper, which is unfortunate. Whe have added the link, https://www.geosci-model-dev-discuss.net/gmd-2017-52/ to the references.

Major comments

R1/1. It is unclear why you need to vary the peat depth in your optimization. You apparently have measurements of the depth (lines 22-23, page 4), so you ought to use that as a non-calibratable value in the model. The high sensitivity of your parameter calibration to the peat depth implies to me that some other factor must be important and not properly resolved in your underlying model (e.g., O2 profiles below the WT depth or rooting profiles). The argument on line 31, page 6 that it is more computationally expensive to run with a deeper peat depth is not sufficient to justify this approach.

A: Peat depth of a wetland is not constant and therefore using a measured value is not as straightforward as is suggested. The peat depth affects in sqHIMMELI both production and transport. The high sensitivity of the parameter tau_C_cato suggest, as discussed, that the total catotelm decomposition rate is relatively constant as it is positively correlated with peat depth.

We changed the model to include the full 4m deep peat layer, by increasing the thickness of the lowest layer, and mention that this value is not an absolute truth but is to be looked at together with the tau_C_cato parameter, because of the variability of peat depth at any given wetland site, and the heterogeneity of the site. In MCMC experiments, 500000 sequential simulations are often needed, and the posterior estimate improves with more simulations, so in our opinion computational efficiency does matter. If the model simulation takes five seconds longer to finish, we end up spending 29 days more on the simulations.

As both referees wondered about the changing discretization level numbers, We drop the part of the study looking into that, and only concentrate on a single experiment.

R1/2. Changing z_exu and Q10 on a yearly basis seems arbitrary. For Q10, I would expect much larger seasonal than interannual variations, yet you ignore that possibility. I think you need to explicitly describe the mechanisms you are proposing for the inter-annual variability of these parameters. The citation to Bergman et al 2000, who noticed change in Q10, "even within a single year", seems to bely your approach.

A: We could have added an additional model for the parameters Q10 and z_exu, but decided that a simpler approach serves us better here as we do not want to overfit the parameters. Bergman reports the following mid-July – late Sept. Q10-values for minerotrophic lawn, which is closest to the Siikaneva site (Laine et al.) whose data is used in the manuscript: 5.9, 3.7, 8.4, 4.0, 7.1, 4.1, 7.0. This suggests that the seasonal variation can be quite irregular at such sites, and in such a situation we opt for the simplest description of the variation. We would like to note here, that for the calibration we feel it is enough to be convinced that there is variation, and the mechanisms are a research question to be tackled in a separate research project. However, we note that these mechanisms are discussed in e.g. (Davidson et al., 2006), and we add a short section about it to our text.

R1/3. Ignoring the temperature sensitivity of CH4 oxidation appear to be a flaw in

your approach. There is substantial evidence that this temperature dependence is even larger than that for CH4 production. Further, since the net surface emission is a small balance of production and oxidation rates, and you explicitly account for the CH4 production temperature sensitivity (eq. (5)), not including Delta(E_R) in your equation (7) seems unreasonable. a. This concern propagates to the last sentence in your abstract. Oxidation often strongly affects the net CH4 emission, which is the measurement you are using to calibrate. If parameters affecting oxidation are 'not identifiable', then it seems unlikely that the production parameters are reasonable.

A: We re-performed the simulations and added Delta(E_R) in the parameters to be optimized. We add the relevant parts to the discussion and reformulate the abstract to reflect the changes. We also added the Delta_E_oxid parameter to the optimization, that regulates the temperature dependence of oxidation of the parameters. It is true that the oxidation and production terms are correlated, and this is something to beb expected. Nonetheless, also information regarding their correlation is important and as such scientifically valuable, as their mutual relationship may still be well defined.

R1/4. On line 21 of page 7, you state that V_R0 affects the rate of temperature dependent HR, but the T dependency is actually governed by Delta(E_R) which is not used in the calibration (Table 2). In general, it is unclear in your section 3.4 how the CH4 production occurs and its relationship with heterotrophic respiration.

A: This is correct, the shape of the temperature response is governed by Delta(E_R). We clarify the functions of the parameters regarding the HR in the text and also discuss the role of Delta(E_R) that was added in the new simulations.

R1/5. In section 4.1, you say the model was linearized, but you did not show whether such a linearization is a reasonable approach. Please provide a quantitative evaluation of how appropriate this linearization

A: This was an error in the text. The linearization was done for the posterior probability density function and not to the model, and was used only in estimating the initial

proposal covariance for MCMC. In the new simulations we do not use this method and hence this part of the text is dropped.

R1/6. Lines 22-25, page 9: Having to restart the model on January 1 of each year because realistic column gas concentrations were otherwise not predicted is a red flag for a problem in the model. The model should be able to run continuously without interruption. If this is a real problem in the model, you should rectify it. Once rectified and described in the paper, restarting each year for computational efficiency and parallelization is reasonable.

A: The concentrations are realistic and stable and the reason for the restarting is just the optimization algorithm, which allows using separate parallel simulations for different years. We wanted the peat column gas concentrations of any year in the optimizations to be in the regime of the parameters to be optimized, but since the variations are small, we actually believe, that this is more than enough. In the end the model will of course be run linearly in a single simulation, and this can be done already now with only trivial changes required to the model.

R1/7. Section 4.2.4 seems to apply that your objective function is only based on annual values, but the text implies that you use the annual values to linearize the model, and then perform the parameter calibration with observed daily CH4 emissions. Please clarify.

A: The annual component was dropped from the new simulations as they had no practical effect to the posterior. The text is updated accordingly.

Smaller comments:

R1/8. Methane is the second most important anthropogenic GHG for warming (don't forget water vapor).

A: This is now mentioned in the text.

R1/9. You describe annual calibration in they abstract, but not the fact that you used a

ROM and then daily fluxes for calibration (as far as I can tell). This approach should be described in the abstract.

A: The annual values were dropped, see R1/7. Also, a reduced order model was not used, and in the updated work it is not used even for covariance estimation.

R1/10. Line 13-15, page 2: cite recent methane model inter-comparisons here: Melton et al., Bohn et al.

A: Citations were added to text.

R1/11. Your assertion (lines 33-34, page 2 to line 4, page 3) that flawed physics representations, numerical errors, and coding errors are good reasons to calibrate a model is shocking. Calibrating a poorly constructed model is a cardinal sin of modeling, although it is regularly done. I think you might re-think the organization of this paragraph.

A: We clarify the text. However, we would like to point out that even excellent models require calibration. Model parameter optimization is effectively inverse modeling, which can improve predictive performance, reveal bottlenecks, and in the best cases provide information for analyzing the physical system. This being said, it is fully true that blindly done model calibration can lead to strange results and a worse model. We reorganize the section to better clarify our views on the topic. We also mention that proper description of the physics is important for the calibration exercise to make sense.

R1/12. Line 5, page 3: Possibly the most mechanistic and realistic terrestrial CH4 model available today is ecosys (Grant, 2002), which you should cite.

A: Citation has been added to the text.

R1/13. Line 8, page 3: define 'multi modality'

A: The text has been clarified in regard to this.

R1/14. Your figures are cited out of order in the text (e.g., figure 11 cited just after figure 2).

A: The figures are now in order.

R1/15. Line 27, page 12: do you mean 'inter-annual variability' instead of 'annual variability'?

A: Yes, fixed.

R1/16. Line 9, page 12: there is no figure 6g.

A: Should have been 6 (b), fixed

R1/17. Does the model calculate the peat temperature? It is not clear from your description which T you are using to estimate your temperature sensitivity. Air T?

A: The model uses any soil temperatures that it is given. In this work we used everywhere measured soil temperatures. We clarify this point further.

R1/18. What happened to a discussion of figure 10?

A: A short discussion was left out and is now added.

R1/19. Where did the NPP come from? Describe in Methods.

A: This is explained in the appendix, but is clarified in the main text now.

Anonymous Referee #2

The objective of this paper is to use observed carbon flux time series in order to optimize parameters of a peatland carbon flux model. In general, this is a timely and important work. However, I found several serious issues with this manuscript including potential flaws in the method that does not allow a publication in the present form.

Please, indicate in abstract and introduction, what is the overall objective of this model e.g. in future applications? Do you want to apply it exclusively for this one peat site and for which question? Do you want to apply it on a continental to global scale, e.g. as part of a land surface scheme? In the latter case, several model assumptions are not useful (effective peat depth, C pool-independent decomposition flux), and a lot of work

on parameter optimization seems to be questionable when only data from one specific site is used.

A: We thank Anonymous Referee #2 for his/her valuable comments.

The HIMMELI model will be used in both stand-alone configurations and as parts of land surface components of regional and global models. We do not intend that the parameter optimization work here would be relevant to any wetland site – rather we look at within-site variation of parameters. The model assumption-related problems are addressed in the points below. We also further clarify the objectives of the research in the abstract and the introduction.

Major comments to sqHIMMELI assumptions:

R2/1 In both aerobic and anaerobic cases, organic matter decay seems to be a constant parameter not depending on substrate availability. When peat depth would have been set constant to the observed site-level value, then this could be valid for the specific site but then we do not learn anything from the parameter optimization procedure for a generally applicable dynamic model.

A: In the model, the organic matter decay (anoxic peat decomposition, Eq. A6) is not a constant parameter but depends also on the amount of peat and its temperature. And of course e.g. oxygen concentration affects the decay rates. We would like to point out for clarity that Eq. 5 in itself does not determine the reaction rate. In addition to moving appendix A into main text (as requesed in referee comment R2/15), we clarify in the text this point.

We re-performed the simulations with a 4m total peat column, which reflects our knowledge of the peat depth at the Siikaneva site (Rinne et al. 2007). Anaerobic respiration of exudates on the other hand does not depend on the peat column thickness. We clarify this in the text.

As peatlands differ widely from site to site, naturally these different types of wetlands

have different parameters controlling the organic matter decay. In the manuscript we look at the decay rates for a single site and as such the parameter values are not directly generalizable everywhere. However, we believe that we could, by looking at data from other sites, calibrate the model for various types of wetlands with e.g. hierarchical Bayesian methods, and this would make it possible to e.g. use the model with land surface sceme. This is work still waiting to be done and beyond the scope of this manuscript.

R2/2 Effective peat depth: This assumption makes no sense at all. Peat depth should be a constant value corresponding to the site observation. See below for related flaws in eq 5.

A: We have changed the peat depth to 4m that roughly represents the peat depth in Siikaneva. We no longer speak about effective peat depth in the manuscript. The different simulations now refer to how deep the fine discretization of the peat column goes. Please see also the answer to R1/1.

R2/3 A2 Anaerobic respiration producing CH4: It seems from eq. A5-A8 that you apply a CH4:CO2 ratio of 1:1 for anaerobic decomposition of root exudates. If so, please make this statement explicit and cite experimental literature showing this ratio.

A: We initially optimized the ratio along with other parameters, but due to covariability with the z_exu parameter without CO2 data, we left this parameter out as only z_exu or the ratio-determining parameter could be determined, and for the first iteration we chose 1:3. However, with CO2 flux data, we are able to constrain the parameter, and therefore we add it to the optimization and to the discussion.We now use data from (Nilsson & Öquist) to set the prior values and explicitly state the final ratios.

R2/4 In section 3 it is also fully unclear if you consider anaerobic CO2 production or not.

A: We do. This is now also clarified in the text.

R2/5 Eq. 5: is tau(cato) the mean residence time at 273.15 K? The unit (y) in Tab 3 is not correct because in eq. 5 you do not multiply with a Cpool.

A: Equation 5 just describes a rate parameter, equation A6 is the actual peat decomposition. When tau has the units of time, A6 then has the time in the denominator which is correct.

R2/6 I expect the Finland peat being frozen with snowpack above over long time periods of the year. What are the effects of <273K soil temperature on aerobic and anaerobic decomposition? What are the effects of soil ice on gas transport and what are the effects of snow on gas transport?

A: Ice and snow slow diffusion of gases into the atmosphere, but this has not been so far implemented in HIMMELI, except for some very preliminary and simple efforts. We tried increasing the resistance of the top soil layer when top soil temperature fell under 0 C but this did not improve the fit / change the results enough for that the change would have been reasonable to keep. We have hence not included descriptions for processes such as diffusion through snow, or release of accumulated gas bubbles under ice in spring time as described by e.g. Mastepanov et al. (2013), Sriskantharajah et al. (2012) – this will be very interesting and will hopefully be done at a later stage.

R2/7 Please include in results and discussions the exudate pool values.

A: We add the exudate pool values and briefly discuss them.

R2/8 Peat depth: Prescribing an effective peat depth will hinder any application of that model in larger dynamic models, such as land surface schemes or DGVMs. Peat depth is no parameter there that you can prescribe but included into the mass balance equations. If you define an effective peat depth then this would mean that you either introduce a fully recalcitrant carbon pool (case peat depth > effective peat) or that you "produce" CH4 and CO2 from non-existing carbon (case peat depth < effective peat depth). That is not a valid and also not useful model assumption.

A: We are not modeling for peat depth changes in this model so far. Of course, the decay in m yˆ-1 can be calculated in a straightforward way, but since we don't know the speed of new peat formation we don't know the change of the peat column in time. This work will be done later. We have changed the peat column depth to reflect the real depth. We have dropped the notion of effective peat depth.

Major comments on the parameter optimization:

R2/9 Tab 5: What is the reason for not including these parameters into MCMC optimization? I generally think that the information content in the data is far too low for an optimization of all model parameters, hence a selection will be useful. However, we need good reasons for such selection, either based on theory or based on a previous sensitivity analysis.

A: The selection was based on a previous preliminary analysis. This is now mentioned in the text. In practice not fixing some of the parameters may lead to ending up in local minima that are unrealistic. However, more importantly, parameters are interlinked via the model processes and in order to constrain the parameters determining the most important processes we fix some of the less important ones. We now explain for all non-included parameters why they have not been included.

R2/10 I assume there is additional CO2 flux data available at the site. It is totally unclear why this data has not been used for constraining in addition to CH4 parameters such as decomposition and transport parameters as well as oxidation parameters

A: We have re-performed the simulations also utilizing CO2 flux data. The setup description, results, and the discussion have been updated.

R2/11 parameter values cannot be transferred to other similar models and even not to HIMMELI because of the peat depth parameter and because of important differences in model formulations: root depth distribution, decomposition parameterization, etc. What is the scientific value of the paper then? Do you plan to use this model version in

future studies and not the HIMMELI model?

A: We plan to use both sqHIMMELI and HIMMELI in future studies. The decomposition parameterization is not part of the HIMMELI model but it can still be used as a source of the anaerobic respiration, which is an input variable for HIMMELI. We believe that the results are transferrable to other models variably: the optimal values of course are not so straightforward, but correlations of the processes more so. HIMMELI is a model more suitable for integration in land surface schemes, whereas sqHIMMELI is a version of the model more designed to be used in stand-alone experiments and settings such as MCMC studies. We will in future studies also integrate features from both model version's development into each other. What model version we use for future studies will depend on the research question at hand.

R2/12 section A4: I do not understand the sentence "Due to coding mistake, the $f_{D,a}$ and $f_{D,w}$ coefficients in the aforementioned equations were set to 0.1 for gases other than $CH_4$ in this work." Why do you set both parameters to 0.1? With a huge pore volume in peat soils I would expect a value of 0.8 or 0.9. If that is a tuning parameter then you should optimize it. These parameters are also not listed in Tab 5. Instead they are part of Fig 2 and this seems to be a real flaw in the procedure?

A: This flaw has been fixed in the new simulations.

R2/13 For clarity, please put units on all parameters in tables and figures or when describing parameters in the text.

A: This has been done.

R2/14 I cannot understand the a posteriori optimized parameter values of tau_exu in the order of magnitude 0.00001 s (tab 3) when range is 3 to 30 days with a prior of 14 days (tab 2). From Fig 3 it seems there is a mistake in units in the table. I have similar problems with units of a posteriori V_0R which seems to be far too high. Zeta_exu seems to be with 0.5 also quite high and it would be good to see some comparison

to literature values if available in the discussion. tau(cato) ranges from 2000 to 20000 years (unit in tab 3 wrong however) depending on peat depth just because the model invalidly does not take the carbon pool into account for calculating the decomposition flux (eq. 5). Then of course, the deeper the peat the more C available the higher you need to have turnover time for the same flux. This is not a valid approach for a dynamic model.

A: For the units, we believe that Table 3 is correct: if tau_exu x 0.00001 = 10, then tau_exu = 10ˆ6, which is around 11.6 days. Same is true for tau_cato and V_0R. The zeta_exu parameter is high because of a non-optimized another parameter f_methane in the model.That parameter value has been added to the optimization.  We would like to note also that the decomposition flux is given by equation A6 – equation 5 gives just the reaction rate constant and if tau_cato has the units of years, the peat decomposition flux given by A6 becomes, integrated over the depth, moles per second per square meter. For the last sentence, we refer to replies to R2/1 and R2/5.

R2/15 A minor comment: I do not find it useful to have some methods description in the main text and some in appendix A but both relate so strong to each other that one understands it only when reading both together.  Please move appendix A into main methods text.

A: We have moved the appendix A into the main methods text.

References:

Erica Davidson, Ivana Janssens, Yiqi Luo: On the variability of respiration in terrestrial ecosystems: moving beyond Q 10, Global Change Biology (2006) 12, 154–164, doi: 10.1111/j.1365-2486.2005.01065.x

Laine AM, Bubier J, Riutta T, Nilsson MB, Moore TR, Vasander H, Tuittila E-S. 2012. Abundance ad composition of plant biomass as potential controls for mire net ecosystem CO2 exchange, Botany Vol 90, 63-74.

M. Mastepanov, C. Sigsgaard, T. Tagesson, L. Ström, M. P. Tamstorf, M. Lund, and T. R. Christensen: Revisiting factors controlling methane emissions from high-Arctic tundra, Biogeosciences, 10, 5139–5158, 2013 doi:10.5194/bg-10-5139-2013

Nilsson & Öquist: Partitioning Litter Mass Loss into Carbon Dioxide and Methane in Peatland Ecosystems, Carbon Cycling in Northern Peatlands, Geophysical Monograph Series 184

Rinne, J., Riutta, T., Pihlatie, M., Aurela, M., Haapanala, S., Tuovinen, J.-P., Tuittila, E.-S., and Vesala, T.: Annual cycle of methane emission from a boreal fen measured by the eddy covariance technique, Tellus B, 59, 449–457, doi:10.1111/j.1600-0889.2007.00261.x, http://dx.doi.org/10.1111/j.1600-0889.2007.00261.x, 2007.

S. Sriskantharajah, R.E. Fisher, D.Lowry, T. Aalto, J. Hatakka, M. Aurela, T. Laurila, A. Lohila, E. Kuitunen: Stable carbon isotope signatures of methane from a Finnish subarctic wetland, Tellus B 2012, 64, 18818, http://dx.doi.org/10.3402

---

## Author Response (AR1)

*Editor comments*

**E1: The main paper must give the model name and version number (or other unique identifier) in the title.**

**AE1:** The final paper contains the model name and version number in the title.

**CE1:** The title is now changed.

*Anonymous Referee #1*

**The authors use a reduced order model representation of a complex numerical wet land methane model with adaptive MCMC to estimate posterior distributions of model parameters. Because wetland methane models are complex and surface CH4 emissions are small differences between large gross fluxes, this type of calibration exercise is difficult and potentially valuable.**

**The authors did a nice job describing their results, given their model setup and assumptions. At this point, however, several assumptions in the approach leave me unconvinced of the reasonableness of the results, which I describe below. Also, the underlying model has not been described in the literature, and citing "Raivonen et al. (2017)" is inappropriate, since it is apparently a paper in preparation.**

We thank the Anonymous Referee #1 for her/his constructive critique and we address the points raised below. Here, we would like to point out that Raivonen et al. Is in public discussion in GMD, and therefore available for consultation. The list of references that we provide at the end of the article has the old reference to "in preparation" paper, which is unfortunate. Whe have added the link, https://www.geosci-model-dev-discuss.net/gmd-2017-52/ to the references.

**Changes to manuscript:** Reference to Raivonen changed in the bibliography; now the entry is correct.

**Major comments**

**R1/1. It is unclear why you need to vary the peat depth in your optimization. You apparently have measurements of the depth (lines 22-23, page 4), so you ought to use that as a non-calibratable value in the model. The high sensitivity of your parameter calibration to the peat depth implies to me that some other factor must be important and not properly resolved in your underlying model (e.g., O2 profiles below the WT depth or rooting profiles). The argument on line 31, page 6 that it is more computationally expensive to run with a deeper peat depth is not sufficient to justify this approach.**

**AR1/1:** Peat depth of a wetland is not constant and therefore using a measured value is not as straightforward as is suggested. The peat depth affects in sqHIMMELI both production and transport. The high sensitivity of the parameter tau_C_cato suggest, as discussed, that the total catotelm decomposition rate is relatively constant as it is positively correlated with peat depth.

We changed the model to include the full 4m deep peat layer, by increasing the thickness of the lowest layer, and mention that this value is not an absolute truth but is to be looked at together with the tau_C_cato parameter, because of the variability of peat depth at any given wetland site, and the heterogeneity of the site. In MCMC experiments, 500000 sequential simulations are often needed, and the posterior estimate improves with more simulations, so in our opinion computational efficiency does matter. If the model simulation takes five seconds longer to finish, we end up spending 29 days more on the simulations.

As both referees wondered about the changing discretization level numbers, We drop the part of the study looking into that, and only concentrate on a single experiment.

**CR1/1** Regarding peat depth: changes in Sec. 3.3 explain this, latexdiff p.8. l. 7-12; also the connection to the tau_C_cato parameter is mentioned there.

**R1/2. Changing z_exu and Q10 on a yearly basis seems arbitrary. For Q10, I would expect much larger seasonal than interannual variations, yet you ignore that possibility. I think you need to explicitly describe the mechanisms you are proposing for the inter-annual variability of these parameters. The citation to Bergman et al 2000, who noticed change in Q10, "even within a single year", seems to bely your approach.**

**AR1/2** We could have added an additional model for the parameters Q10 and z_exu, but decided that a simpler approach serves us better here as we do not want to overfit the parameters. Bergman reports the following mid-July – late Sept. Q10-values for minerotrophic lawn, which is closest to the Siikaneva site (Laine et al.) whose data is used in the manuscript: 5.9, 3.7, 8.4, 4.0, 7.1, 4.1, 7.0. This suggests that the seasonal variation can be quite irregular at such sites, and in such a situation we opt for the simplest description of the variation. We would like to note here, that for the calibration we feel it is enough to be convinced that there is variation, and the mechanisms are a research question to be tackled in a separate research project. However, we note that these mechanisms are discussed in e.g. (Davidson et al., 2006), and we add a short section about it to our text.

**CR1/2** Text added to Sec. 3.4, latexdiff p. 8 l.18-21

**R1/3. Ignoring the temperature sensitivity of CH4 oxidation appear to be a flaw in your approach. There is substantial evidence that this temperature dependence is even larger than that for CH4 production. Further, since the net surface emission is a small balance of production and oxidation rates, and you explicitly account for the CH4 production temperature sensitivity (eq. (5)), not including Delta(E_R) in your equation (7) seems unreasonable. a. This concern propagates to the last sentence in your abstract. Oxidation often strongly affects the net CH4 emission, which is the measurement you are using to calibrate. If parameters affecting oxidation are 'not identifiable', then it seems unlikely that the production parameters are reasonable.**

**AR1/3** We re-performed the simulations and added Delta(E_R) in the parameters to be optimized. We add the relevant parts to the discussion and reformulate the abstract to reflect the changes. We also added the Delta_E_oxid parameter to the optimization, that regulates the temperature dependence of oxidation of the parameters. It is true that the oxidation and production terms are correlated, and this is something to beb expected. Nonetheless, also information regarding their correlation is important and as such scientifically valuable, as their mutual relationship may still be well defined.

**CR1/3** The parametrers are added to Sec. 3.4, parameters no 7 and 9, latexdiff p.9 l.13 and 17. Also, they are added to the results, and tables at least to the following spots: prior information Table 3, Figs 3,4,6, and results in Table 4. Also latexdiff p.18 around l.5, p.22 l. 22, p.23 l.7, p.26 l.1, p.26 l.5.

**R1/4. On line 21 of page 7, you state that V_R0 affects the rate of temperature dependent HR, but the T dependency is actually governed by Delta(E_R) which is not used in the calibration (Table 2). In general, it is unclear in your section 3.4 how the CH4 production occurs and its relationship with heterotrophic respiration.**

**AR1/4**This is correct, the shape of the temperature response is governed by Delta(E_R). We clarify the functions of the parameters regarding the HR in the text and also discuss the role of Delta(E_R) that was added in the new simulations.

**CR1/4** This is actually covered by the description of parameter V_R0, parameter no. 6, latexdiff p.9. The short text there explains roles of both of those parameters, and the equation where they function. The parameter DeltaE_R is now part of optimization, see response CR1/3.

**R1/5. In section 4.1, you say the model was linearized, but you did not show whether such a linearization is a reasonable approach. Please provide a quantitative evaluation of how appropriate this linearization**

**AR1/5**This was an error in the text. The linearization was done for the posterior probability density function and not to the model, and was used only in estimating the initial proposal covariance for MCMC. In the new simulations we do not use this method and hence this part of the text is dropped.

**CR1/5** Sec. 4, latexdiff p. 13 l.20-21, removed text.

**R1/6. Lines 22-25, page 9: Having to restart the model on January 1 of each year because realistic column gas concentrations were otherwise not predicted is a red flag for a problem in the model. The model should be able to run continuously without interruption. If this is a real problem in the model, you should rectify it. Once rectified and described in the paper, restarting each year for computational efficiency and parallelization is reasonable.**

**AR1/6**The concentrations are realistic and stable and the reason for the restarting is just the optimization algorithm, which allows using separate parallel simulations for different years. We wanted the peat column gas concentrations of any year in the optimizations to be in the regime of the parameters to be optimized, but since the variations are small, we actually believe, that this is more than enough. In the end the model will of course be run linearly in a single simulation, and this can be done already now with only trivial changes required to the model.

**CR1/6** We believe this is adequately explained in the text, but changed the text a little, latexdiff. p.14 l. 17.

**R1/7. Section 4.2.4 seems to apply that your objective function is only based on annual values, but the text implies that you use the annual values to linearize the model, and then perform the parameter calibration with observed daily CH4 emissions. Please clarify.**

**AR1/7**The annual component was dropped from the new simulations as they had no practical effect to the posterior. The text is updated accordingly.

**CR1/7** See CR1/5, also latexdiff p.14 l.25. Additionally, removed Annual CH4 fluxes section, latexdiff p.15 l. 11-20

**Smaller comments:**

**R1/8. Methane is the second most important anthropogenic GHG for warming (don't forget water vapor).**

**AR1/8** This is now mentioned in the text.

**CR1/8** p.2 l.11-12

**R1/9. You describe annual calibration in they abstract, but not the fact that you used a ROM and then daily fluxes for calibration (as far as I can tell). This approach should be described in the abstract.**

**AR1/9** The annual values were dropped, see R1/7. Also, a reduced order model was not used, and in the updated work it is not used even for covariance estimation.

**CR1/9** No additional text changes needed as no ROM  nor annual estimates were in the end used

**R1/10. Line 13-15, page 2: cite recent methane model inter-comparisons here: Melton et al., Bohn et al.**

**AR1/10** Citations were added to text.

**CR1/10** Latexdiff: mention added in intro, p. 2/l. 27

**R1/11. Your assertion (lines 33-34, page 2 to line 4, page 3) that flawed physics representations, numerical errors, and coding errors are good reasons to calibrate a model is shocking. Calibrating a poorly constructed model is a cardinal sin of modeling, although it is regularly done. I think you might re-think the organization of this paragraph.**

**AR1/11** We clarify the text. However, we would like to point out that even excellent models require calibration. Model parameter optimization is effectively inverse modeling, which can improve predictive performance, reveal bottlenecks, and in the best cases provide information for analyzing the physical system. This being said, it is fully true that blindly done model calibration can lead to strange results and a worse model. We reorganize the section to better clarify our views on the topic. We also mention that proper description of the physics is important for the calibration exercise to make sense.

 **CR1/11** latexdiff p.3, l.13-20

**R1/12. Line 5, page 3: Possibly the most mechanistic and realistic terrestrial CH4 model available today is ecosys (Grant, 2002), which you should cite.**

**AR1/12** Citation has been added to the text.

**CR1/12** Citation added, latexdiff p.3 l.22

**R1/13. Line 8, page 3: define 'multi modality'**

**AR1/13** The text has been clarified in regard to this.

**CR1/13** latexdiff p.4 l. 26-27., text clarified.

**R1/14. Your figures are cited out of order in the text (e.g., figure 11 cited just after figure 2).**

**AR1/14** The figures are now in order.

**CR1/14** Figures and Tables are in order of appearance, and hence they are also in order in the text.

**R1/15. Line 27, page 12: do you mean 'inter-annual variability' instead of 'annual variability'?**

**AR1/15** Yes, fixed.

**CR1/15** latexdiff p.18 l. 21

**R1/16. Line 9, page 12: there is no figure 6g.**

**AR1/16** Should have been 6 (b), fixed

**CR1/16** latexdiff p.19 l. 14;  (due to ordering changes of the figures and changes in the figures, the numbers are now different, and the old numbers are wrong; the cross validation figures are now 7b and 8)

**R1/17. Does the model calculate the peat temperature? It is not clear from your description which T you are using to estimate your temperature sensitivity. Air T?**

**AR1/17**The model uses any soil temperatures that it is given. In this work we used everywhere measured soil temperatures. We clarify this point further.

**CR1/17** clarified that it was measurement data  that was used. latexdiff p.5 l.31-32

**R1/18. What happened to a discussion of figure 10?**

**AR1/18**A short discussion was left out and is now added.

**CR1/18** This figure is now discussed: latexdiff p.21 l.24-28. The figure was already touched upon earlier, this section is latexdiff p.25 l.20-25.

**R1/19. Where did the NPP come from? Describe in Methods.**

**AR1/19**This is explained in the appendix, but is clarified in the main text now.

**CR1/19** Regression modeling of NPP is now mentioned in methods (data description section), latexdiff p.6 l.2-3. The detailed description is still left to the Appendix E.

**Anonymous Referee #2**

**The objective of this paper is to use observed carbon flux time series in order to optimize parameters of a peatland carbon flux model. In general, this is a timely and important work. However, I found several serious issues with this manuscript including potential flaws in the method that does not allow a publication in the present form.**

**Please, indicate in abstract and introduction, what is the overall objective of this model e.g. in future applications? Do you want to apply it exclusively for this one peat site and for which question? Do you want to apply it on a continental to global scale, e.g. as part of a land surface scheme? In the latter case, several model assumptions are not useful (effective peat depth, C pool-independent decomposition flux), and a lot of work on parameter optimization seems to be questionable when only data from one specific site is used.**

The HIMMELI model will be used in both stand-alone configurations and as parts of land surface components of regional and global models. We do not intend that the parameter optimization work here would be relevant to any wetland site – rather we look at within-site variation of parameters. The model assumption-related problems are addressed in the points below. We also further clarify the objectives of the research in the abstract and the introduction.

**Changes to manuscript:** Main results clarified in abstract and intro, latexdiff p.1 l. 11-17. Applicability, p.1 l. 7-9., p.3 l.11-13.

*Major comments to sqHIMMELI assumptions:*

**R2/1 In both aerobic and anaerobic cases, organic matter decay seems to be a constant parameter not depending on substrate availability. When peat depth would have been set constant to the observed site-level value, then this could be valid for the specific site but then we do not learn anything from the parameter optimization procedure for a generally applicable dynamic model.**

**AR2/1** In the model, the organic matter decay (anoxic peat decomposition, Eq. A6) is not a constant parameter but depends also on the amount of peat and its temperature. And of course e.g. oxygen concentration affects the decay rates. We would like to point out for clarity that Eq. 5 in itself does not determine the reaction rate. In addition to moving appendix A into main text (as requesed in referee comment R2/15), we clarify in the text this point.

We re-performed the simulations with a 4m total peat column, which reflects our knowledge of the peat depth at the Siikaneva site (Rinne et al. 2007). Anaerobic respiration of exudates on the other hand does not depend on the peat column thickness. We clarify this in the text.

As peatlands differ widely from site to site, naturally these different types of wetlands have different parameters controlling the organic matter decay. In the manuscript we look at the decay rates for a single site and as such the parameter values are not directly generalizable everywhere. However, we believe that we could, by looking at data from other sites, calibrate the model for various types of wetlands with e.g. hierarchical Bayesian methods, and this would make it possible to e.g. use the model with land surface sceme. This is work still waiting to be done and beyond the scope of this manuscript.

**CR2/1** Change of peat depth mentioned: latexdiff Sec. 3 p. 6 l. 21. For peat decay dependence, latexdiff p. 11, l.21-22.
(In fact we ended up in the end using 85% of the maximum depth of 4 meters, as the reported depth was 2-4 meters.)

**R2/2 Effective peat depth: This assumption makes no sense at all. Peat depth should be a constant value corresponding to the site observation. See below for related flaws in eq 5.**

**AR2/2** We have changed the peat depth to 4m that roughly represents the peat depth in Siikaneva. We no longer speak about effective peat depth in the manuscript. The different simulations now refer to how deep the fine discretization of the peat column goes. Please see also the answer to R1/1.

**CR2/2** Only one discretization was used in the end. Text changed everywhere to reflect that we now use an approximation of the actual peat depth, e.g. latexdiff p.8 l.7-11. All figures and tables that touched upon different peat depths are updated (new numbers): Tables 4 and 5, and Figures 4,6,7,8,10,12,13,

**R2/3 A2 Anaerobic respiration producing CH4: It seems from eq. A5-A8 that you apply a CH4:CO2 ratio of 1:1 for anaerobic decomposition of root exudates. If so, please make this statement explicit and cite experimental literature showing this ratio.**

**AR2/3**We initially optimized the ratio along with other parameters, but due to covariability with the z_exu parameter without CO2 data, we left this parameter out as only z_exu or the ratio-determining parameter could be determined, and for the first iteration we chose 1:3. However, with CO2 flux data, we are able to constrain the parameter, and therefore we add it to the optimization and to the discussion.We now use data from (Nilsson & Öquist) to set the prior values and explicitly state the final ratios.

**CR2/3** f_exu^CH4 parameter added, latexdiff p.9 l.6, p.11 l.20, p.17, l.31-33, p.21 l.20, p.26 l.19, also Figs 3,4,6, Tables 3 and 4.

**R2/4 In section 3 it is also fully unclear if you consider anaerobic CO2 production or not.**

**AR2/4**We do. This is now also clarified in the text.

**CR2/4** Mentioned in Sec. 3 first paragraph; latexdiff p. 6 l. 11

**R2/5 Eq. 5: is tau(cato) the mean residence time at 273.15 K? The unit (y) in Tab 3 is not correct because in eq. 5 you do not multiply with a Cpool.**

**AR2/5**Equation 5 just describes a rate parameter, equation A6 is the actual peat decomposition. When tau has the units of time, A6 then has the time in the denominator which is correct.

**CR2/5** We changed the presentation of the units and multiplicative factors in Table 4 (new number) to be more intuitive, even though we believe that it was also correct previously. See Table 4.

**R2/6 I expect the Finland peat being frozen with snowpack above over long time periods of the year. What are the effects of <273K soil temperature on aerobic and anaerobic decomposition? What are the effects of soil ice on gas transport and what are the effects of snow on gas transport?**

**AR2/6**Ice and snow slow diffusion of gases into the atmosphere, but this has not been so far implemented in HIMMELI, except for some very preliminary and simple efforts. We tried increasing the resistance of the top soil layer when top soil temperature fell under 0 C but this did not improve the fit / change the results enough for that the change would have been reasonable to keep. We have hence not included descriptions for processes such as diffusion through snow, or release of accumulated gas bubbles under ice in spring time  as described by e.g. Mastepanov et al. (2013), Sriskantharajah et al. (2012)  – this will be very interesting and will hopefully be done at a later stage.

**CR2/6** References and explanations added, Sec. 3, latexdiff p.6 l. 26-28

**R2/7 Please include in results and discussions the exudate pool values.**

**AR2/7**We add the exudate pool values and briefly discuss them.

**CR2/7** The exudate pool is added to Fig. 9 and there is a short description of it on p. 22 l. 1-3 in the latexdiff.

**R2/8 Peat depth: Prescribing an effective peat depth will hinder any application of that model in larger dynamic models, such as land surface schemes or DGVMs. Peat depth is no parameter there that you can prescribe but included into the mass balance equations. If you define an effective peat depth then this would mean that you either introduce a fully recalcitrant carbon pool (case peat depth > effective peat) or that you "produce" CH4 and CO2 from non-existing carbon (case peat depth < effective peat depth). That is not a valid and also not useful model assumption.**

**AR2/8**We are not modeling for peat depth changes in this model so far. Of course, the decay in m y^-1 can be calculated in a straightforward way, but since we don't know the speed of new peat formation we don't know the change of the peat column in time. This work will be done later. We have changed the peat column depth to reflect the real depth. We have dropped the notion of effective peat depth.

**CR2/8** Same changes apply as in CR1/1, CR2/1, CR2/2

*Major comments on the parameter optimization:*

**R2/9 Tab 5: What is the reason for not including these parameters into MCMC optimization? I generally think that the information content in the data is far too low for an optimization of all model parameters, hence a selection will be useful. However, we need good reasons for such selection, either based on theory or based on a previous sensitivity analysis.**

**AR2/9**The selection was based on a previous preliminary analysis. This is now mentioned in the text. In practice not fixing some of the parameters may lead to ending up in local minima that are unrealistic. However, more importantly, parameters are interlinked via the model processes and in order to constrain the parameters determining the most important processes we fix some of the less important ones. We now explain for all non-included parameters why they have not been included.

**CR2/9.** Latexdiff p. 8 / l.21-22, and Table 2, p. 49 in latexdiff / p. 38 in the manuscript and its caption.

**R2/10 I assume there is additional CO2 flux data available at the site. It is totally unclear why this data has not been used for constraining in addition to CH4 parameters such as decomposition and transport parameters as well as oxidation parameters**

**AR2/10**We have reperformed the simulations also utilizing CO2 flux data. The setup description, results, and the discussion have been updated.

**CR2/10** The changes are wide-ranging because also methods had to be updated, including adding a further resampling step. Latexdiff p.1 l.4, p.4 l. 26, p.5 l.2-29, p.14, l.23-31, p.15, l.3-6 and 14-17, Eq. 24, p.16 l. 5, p. 19 l.3-5, 20-21, 31-33, p.30, l.7-12, 17, p.33 l.26, p. 34 l. 19-24. For the resampling, a section was added, p.35, l. 1- p.36, l.7. and p.38 l.13-20. The section (latexdiff) p. 39 l.21- p. 40 l. 5 were dropped as obsolete.

**R2/11 parameter values cannot be transferred to other similar models and even not to HIMMELI because of the peat depth parameter and because of important differences in model formulations: root depth distribution, decomposition parameterization, etc. What is the scientific value of the paper then? Do you plan to use this model version in future studies and not the HIMMELI model?**

**AR2/11**We plan to use both sqHIMMELI and HIMMELI in future studies. The decomposition parameterization is not part of the HIMMELI model but it can still be used as a source of the anaerobic respiration, which is an input variable for HIMMELI. We believe that the results are transferrable to other models variably: the optimal values of course are not so straightforward, but correlations of the processes more so. HIMMELI is a model more suitable for integration in land surface schemes, whereas sqHIMMELI is a version of the model more designed to be used in stand-alone experiments and settings such as MCMC studies. We will in future studies also integrate features from both model version's development into each other. What model version we use for future studies will depend on the research question at hand.

**CR2/11** This was clarified as indicated  in changes to manuscript in response to general comments by referee #2: this document, top of p. 6. In our view no other changes to the manuscript are needed in addition to those mentioned there. (copied here: *"Main results clarified in abstract and intro, latexdiff p.1 l. 11-17. Applicability, p.1 l. 7-9., p.3 l.11-13.2"*)

**R2/12 section A4: I do not understand the sentence "Due to coding mistake, the fD,a and fD,w coefficients in the aforementioned equations were set to 0.1 for gases other than CH4 in this work." Why do you set both parameters to 0.1? With a huge pore volume in peat soils I would expect a value of 0.8 or 0.9. If that is a tuning parameter then you should optimize it. These parameters are also not listed in Tab 5. Instead they are part of Fig 2 and this seems to be a real flaw in the procedure?**

**AR2/12**This flaw has been fixed in the new simulations.

**CR2/12** p. 33 l.9 sentence removed. (Due to moving the appendix to the main text, latexdiff gets confused and this change is not properly shown in the diff.) The new corresponding place is  (latexdiff) p. 12 l.23.

**R2/13 For clarity, please put units on all parameters in tables and figures or when describing parameters in the text.**

**AR2/13**This has been done.

**CR2/13** Changes all over in text, including the tables and figures. Units were added in parentheses. Manuscript  Figs 3,6,13, and now Tables 1-4 have also units.

**R2/14 I cannot understand the a posteriori optimized parameter values of tau_exu in the order of magnitude 0.00001 s (tab 3) when range is 3 to 30 days with a prior of 14 days (tab 2). From Fig 3 it seems there is a mistake in units in the table. I have similar problems with units of a posteriori V_0R which seems to be far too high. Zeta_exu seems to be with 0.5 also quite high and it would be good to see some comparison to literature values if available in the discussion. tau(cato) ranges from 2000 to 20000 years (unit in tab 3 wrong however) depending on peat depth just because the model invalidly does not take the carbon pool into account for calculating the decomposition flux (eq. 5).**

**Then of course, the deeper the peat the more C available the higher you need to have turnover time for the same flux. This is not a valid approach for a dynamic model.**

**AR2/14**For the units, we believe that Table 3 is correct: if tau_exu x 0.00001 = 10, then tau_exu = 10^6, which is around 11.6 days. Same is true for tau_cato and V_0R. The zeta_exu parameter is high because of a non-optimized another parameter f_methane in the model.That parameter value has been added to the optimization. We would like to note also that the decomposition flux is given by equation A6 – equation 5 gives just the reaction rate constant and if tau_cato has the units of years, the peat decomposition flux given by A6 becomes, integrated over the depth, moles per second per square meter. For the last sentence, we refer to replies to R2/1 and R2/5.

**CR2/14** We changed the way the units and the magnitudes were presented, and even though we believe that it was also previously correct, we feel it is now more intuitive. See "parameter"-column in Table 4 in the manuscript. (latexdiff had serious difficulty formatting the differences in the table).

**R2/15 A minor comment: I do not find it useful to have some methods description in the main text and some in appendix A but both relate so strong to each other that one understands it only when reading both together. Please move appendix A into main methods text.**

**AR2/15**We have moved the appendix A into the main methods text.

**CR2/15** Appendix A in the old version moved to be Sec. 3.5 in the new version.

[revised manuscript text omitted]

where in Eq. 15 $\nu$ is the decay rate of root exudates from Eq. 3, $\eta$ is an oxygen inhibition parameter, $C_{O_2}(z)$ is the oxygen concentration at depth $z$, and $\pi(z)$ is the normalized proportion of the total anaerobic root mass, also at depth $z$, given in an unnormalized form in Eq. 2. The parameter $f_{CH_4}^{exu}$ is a constant determining what fraction of root exudates in anaerobic conditions will turn into $CH_4$. Equation 15 is only used below the water table. In Eq. 16, $g_m^{Q_{10}}$ is the proportion of the anaerobic peat decomposition process producing $CH_4$, $\rho_{cato}$ is the peat density in the catotelm, $f_{C_{cato}}$ is the fraction of carbon in catotelm peat, and $M_C$ is mass of carbon. The parameter $k_{cato} = Q_{10}^{\frac{(T-273.15)}{10}}/\tau_{cato}$ is described in Eq. 5, and is zero above water table.

The equations for $CO_2$ are similar:

$$R_{CO_2}^{exu}(z) = \nu\pi(z) - R_{CH_4}^{exu}(z) R_{CO_2}^{peat}(z) = (1-g_{CH_4}^{Q_{10}})k_{cato}(z)\frac{\rho_{cato}f_{C_{cato}}}{M_C}, \tag{A3}$$

and the meanings of the symbols are analogous to the ones in equations for $CH_4$.

**A1  Peat respiration and methane oxidation**

Peat respiration (aerobic respiration) is described with an equation of the Michaelis-Menten form

$$R_{aerob}^{peat}(z) = V_R(z)\frac{\alpha C_{O_2}^x(z)}{K_R + C_{O_2}^x(z)}, \tag{A4}$$

where $\alpha$ is a dimensionless Henry solubility constant for oxygen above the water table, and one below it, see Tang et al. (2010). The factor $C_{O_2}^x$ refers to $C_{O_2}^w$ below the water table, and to $C_{O_2}^a$ above it. Here $w$ In section 4.2.1 we described the error models as AR(1) / ARMA(2,1) models where the residuals are Laplace-distributed. Intuitively these models can be thought of as characterizing the "inertia" or "memory" in the model-observation discrepancy. Formally the observation equation for our statistical inference problem can be written as

$$\underset{\sim}{\boldsymbol{y}}_t = \underset{\sim}{\boldsymbol{x}}_t + \underset{\sim}{\boldsymbol{r}}_t^* \tag{A5}$$

[revised manuscript text omitted]

---

## Author Response (AR2)

Author responses to referee comments, 2018-02-15

We thank the anonymous referee once again for kindly providing the constructive critique. We provide the answers to the comments below.

**Comment 1**: please include a comparison to CO2 data from the eddy tower

**Response 1**: We have added a figure showing the CO2 flux from the eddy tower and the flux predicted by  the model, and we added a short description of this to the text.

**Changes to MS 1**: Added Fig. 10 and its caption on p. 50, and a short section in the text, p. 17 l. 3-6.

**Comment 2**: about CH4:CO2 ratio: Why is that now different for exudates and bulk organic matter. You are using 0.4 for bulk OM and 0.7 for exudates. I cannot see a scientific explanation for this huge difference. Please, discuss this difference in teh MS secition discussion and explore what are the consequences for a future application of the model.

**Response 2**: The anaerobic respiration is modeled according to two separate processes that are assumed to be different, and therefore there are two different parameters controlling the processes. However, it is true that the difference ends up being quite large after the model calibration. We have added a paragraph noting this discrepancy, and suggesting that in future studies that parameter could also be added to the set of optimized parameters. We also mention that due to the posterior structure this has less effect on the fluxes than it might look based on looking at the difference between the two numbers, but that the balance of the processes could be slightly affected.

**Changes to MS 2**: Added paragraph on p. 18, l. 15-27.

[revised manuscript text omitted]